# Nonparametric Teaching for Multiple Learners

**Chen Zhang**[1]  **Xiaofeng Cao**[1,✉]  **Weiyang Liu**[2,3]  **Ivor W. Tsang**[4]  **James T. Kwok**[5]

[1]School of Artificial Intelligence, Jilin University, China
[2]Max Planck Institute for Intelligent Systems, Germany, [3]University of Cambridge, UK
[4]CFAR and IHPC, Agency for Science, Technology and Research (A*STAR), Singapore
[5]Hong Kong University of Science and Technology, Hong Kong, China
u3567831@connect.hku.hk,xiaofengcao@jlu.edu.cn,wl396@cam.ac.uk
ivor_tsang@cfar.a-star.edu.sg,jamesk@cse.ust.hk

## Abstract

We study the problem of teaching multiple learners simultaneously in the nonparametric iterative teaching setting, where the teacher iteratively provides examples to the learner for accelerating the acquisition of a target concept. This problem is motivated by the gap between current single-learner teaching setting and the real-world scenario of human instruction where a teacher typically imparts knowledge to multiple students. Under the new problem formulation, we introduce a novel framework – Multi-learner Nonparametric Teaching (MINT). In MINT, the teacher aims to instruct multiple learners, with each learner focusing on learning a scalar-valued target model. To achieve this, we frame the problem as teaching a vector-valued target model and extend the target model space from a scalar-valued reproducing kernel Hilbert space used in single-learner scenarios to a vector-valued space. Furthermore, we demonstrate that MINT offers significant teaching speedup over repeated single-learner teaching, particularly when the multiple learners can communicate with each other. Lastly, we conduct extensive experiments to validate the practicality and efficiency of MINT.

## 1   Introduction

Machine teaching [81, 83] considers the problem of how to design the most effective teaching set, typically with the smallest amount of (teaching) examples possible, to facilitate rapid learning of the target models by learners based on these examples. It can be thought of as an inverse problem of machine learning, in the sense that the student aims to learn a target model on a given dataset, while the teacher constructs such a (minimal) dataset. Machine teaching has many applications in computer vision [67, 68], crowd sourcing [59, 60, 78, 79] and cyber security [2, 3, 39, 53].

Roughly speaking, machine teaching can be carried out in a batch [80, 81, 33] or iterative [36, 37, 38, 52] fashion, depending on how teachers and learners interact with each other. Batch teaching focuses on single-round interaction, that is, the most representative and effective teaching dataset are designed to be fed to the learner in one shot. After that, the leaner solely and assiduously learns a target model from this dataset without further interaction. With practical considerations, iterative teaching extends such a single-round mode to a multi-round one. It studies the case where the teacher feeds examples based on learners' status (current learnt models) round by round, such that the learner can converge to a target model within fewer rounds. The minimal count of such rounds (or iterations) is referred to as *iterative teaching dimension* [36, 37].

Considering that previous works on iterative machine teaching usually limit target models in a parameterized family, that is, assuming the target model can be represented by some parameters, nonparametric iterative machine teaching [73] extends such single family to a general nonparametric one. This allows multiple possibility of the target model family. Specifically, by formulating nonparametric teaching in a reproducing kernel Hilbert space (RKHS), [73] introduce various families of target models associated with kernels, *e.g.*, Gaussian and Laplacian kernels in RKHS.

37th Conference on Neural Information Processing Systems (NeurIPS 2023).

However, existing nonparametric teaching only focuses on the single-learner setting (*i.e.*, teaching a scalar-valued target model or function to a single learner), and it is computationally inefficient to carry out this same single-learner teaching repeatedly for the multi-learner scenario, where the teacher needs to teach numerous scalar-valued target functions to multiple learners and a single learner can only learn one. For example, when taking a colored picture with three (RGB) channels as a multi-learner target function[1] (*e.g.*, [62, 20]), the iteration number of repeatedly performing single-learner teaching for each channel is intuitively triple that of carrying out them simultaneously. Another example is the scenario where the single-learner target model having a large input space. Such a model can be divided into multiple smaller ones with an input space of appropriate size, and these smaller models can be formulated together into a multi-learner target model

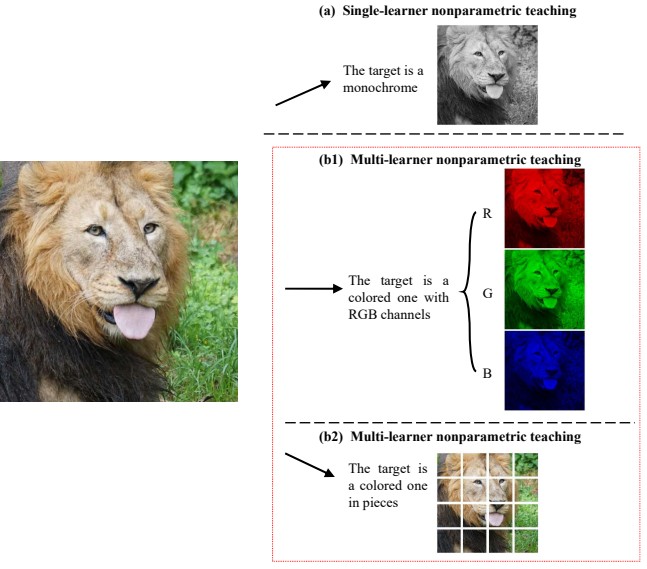

**(a) Single-learner nonparametric teaching**

The target is a monochrome

**(b1) Multi-learner nonparametric teaching**

The target is a colored one with RGB channels

R

G

B

**(b2) Multi-learner nonparametric teaching**

The target is a colored one in pieces

Figure 1: Comparison between the single-learner teaching and MINT. (a) In order to facilitate single-learner teaching, it is imperative to transform a colored image into a grayscale format. (b) MINT allows for the simultaneous teaching of three scalar-valued target models, which are three (RGB) channels of a colored image. (b2) Partitioning a single image into multiple pieces and teaching them concurrently is also considered as a form of MINT.

[7, 64, 66, 61, 74]. Concretely, one can divide a single high-revolution picture into multiple sub-regions, and the original single-learner target function will become a multi-learner one. These examples motivate us to study a generalized framework, called multi-learner nonparametric teaching (MINT), where a vector-valued target model (instead of a scalar-valued model) is being taught. A comparison between single-learner teaching and MINT is illustrated in Figure 1.

It is therefore of great significance to generalize the recent single-learner nonparametric teaching [73] to MINT [11, 14, 77]. MINT is guided by the insight that repeatedly (or sequentially) teaching multiple scalar-valued target functions can be viewed as teaching a vector-valued target function. The theoretical motivation comes from the well-developed results of kernels for vector-valued functions [56, 44, 21], an important approach to deal with multiple data sources. This inspires us to formulate MINT as a teaching problem of a vector-valued target function, where sequentially teaching multiple scalar-valued functions (for single-learner teaching) becomes a special case of teaching a vector-valued function [43, 21, 14]. We emphasize that, compared to the case where a single learner is learning a vector-valued target function, the multi-learner setting offers a general framework that can be generalized to more complicated scenarios, *e.g.*, learners operate within different feature spaces, and learners are able to communicate with each other. We summarize our major contributions below:

- By analyzing general vector-valued RKHS [10, 45, 4], we study the multi-learner nonparametric teaching (MINT), where the teacher selects examples based on a vector-valued target function (each component of the vector-valued function is a scalar-valued function for a single learner)[2], such that multiple learners can learn their own target models simultaneously.

- By enabling the communication among multiple learners, learners can update themselves with a linear combination of current learnt functions of all learners [23, 12].We study a communicated MINT where the teacher not only selects examples but also injects the guidance of communication.

- Under mild assumptions, we characterize the efficiency of our multi-learner generalization of nonparametric teaching. More importantly, we also empirically demonstrate its efficiency.

---

[1]Each channel in a colored picture can be viewed as a single-learner (scalar-valued) target function whose inputs and outputs are the pixel location and values, respectively [73].

[2]When components of it are highly correlated with each other, the teacher in each iteration can select a teaching set based on one component, and use it to teach all learners simultaneously.

## 2 Related Works

**Machine teaching**. Recently, there has been a surge of interest in the field of machine teaching, see [81, 83, 36, 37, 67] and references therein. Batch machine teaching has examined the behaviors of various types of learners, including linear learners [33], forgetful learners [27, 37], version space learners [13, 65], hypothesis learner [40] and reinforcement learners [29, 76]. Further, by extending the single-round teaching mode to a multi-round one, iterative teaching has attracted growing attention in recent studies [36, 37, 40, 50, 32, 38, 69, 52, 73]. Specifically, [38] focuses on label synthesis teaching, while [52] proposes generative teaching. Additionally, [73] relaxes the parametric assumption on target models and generalizes the previous iterative teaching to a nonparametric iterative one [30, 51]. In contrast to previous works that mainly concentrate on the single-learner teaching, this work aims to address a more practical task – teaching multi-learner (vector-valued) target models. In this regard, the realistic practical scenario, classroom teaching [82, 71], is highly relevant, where it examines multiple learners by partitioning them into groups in batch and iterative setting, respectively. However, their works are also limited to the parametric setting, and their methods therefore are not immediately generalizable to nonparametric situations. In contrast, our work investigates multi-learner teaching in the nonparametric setting.

**Multi-task functional optimization**. Functional optimization [58, 6, 58, 22, 84, 63, 75] is a fundamental and significant task in various fields, such as variational inference [35, 34], barycenter problem [57, 70], and Residual Networks [48, 49, 26]. It involves mapping from input to output without having pre-defined parameters, optimized over a more general function space such as the reproducing kernel Hilbert space (RKHS), Sobolev space [1, 46], and Fréchet space [47]. Notably, the functional gradient descent algorithm has been studied extensively for functional optimization in RKHS due to its regular properties [41, 42, 16, 35, 46, 5, 57]. Meanwhile, modeling in RKHS of vector-valued functions [56, 21, 44, 10, 9, 14, 19, 45] is an important approach to handle multi-task problem. Specifically, [21, 9, 4] focus on the analysis of the kernel and [28, 12] study multi-task versions of online mirror descent, which displays the similarity to multi-learner teaching in the sense of simultaneous execution. These theoretical and empirical works motivate us to extend single-learner teaching to a multi-learner one by analyzing functional gradient in vector-valued RKHS.

## 3 Background

**Notation**. Let $\mathcal{X} \subseteq \mathbb{R}^n$ be a $n$ dimensional input (*i.e.*, feature) space and $\mathcal{Y} \subseteq \mathbb{R}$ be a output (*i.e.*, label) space. By $\mathcal{X}^d = \mathcal{X}_1 \times \cdots \times \mathcal{X}_d \subseteq \mathbb{R}^{n \times d}$ we denote a $d$-learner input space, and let $\mathcal{Y}^d = \mathcal{Y}_1 \times \cdots \times \mathcal{Y}_d \subseteq \mathbb{R}^d$ be a $d$-learner output space. Let a $d$-dimensional column vector with $a_i$, entries indexed by $i \in \mathbb{N}_d$ ( $\mathbb{N}_k := \{1, \cdots, k\}$), be $[a_i]^d = (a_1, \cdots, a_d)^T$ (we may denote it by $\boldsymbol{a}$ for simplicity), and a 1-vector of size $d$ be $[1]^d = (1, \cdots, 1)^T \in \mathbb{R}^d$. By $M_{(i,\cdot)}$ we denote $i$-th row vector of a matrix $M$, and let $M_{(\cdot,i)}$ be its $i$-th column vector. A $d$-learner teaching sequence in size $d \times k$ is a collection of examples, notated as $\mathcal{D} = \{(x_{i,j}, y_{i,j}) \in \mathcal{X} \times \mathcal{Y}\}$[3] with the learner index $i \in \mathbb{N}_d$ and the example index $j \in \mathbb{N}_k$. We notate the collection of such teaching sequence candidates by $\mathbb{D}^d \ni \mathcal{D}$, which is referred to as the knowledge domain of the teacher [36].

Let $K(x, x') : \mathcal{X} \times \mathcal{X} \mapsto \mathbb{R}$ be a scalar-valued positive definite kernel function, which can be equivalently notated by $K(x, x') = K_x(x') = K_{x'}(x)$, and one can abbreviate $K_x(\cdot)$ by $K_x$. The scalar-valued reproducing kernel Hilbert space (RKHS) $\mathcal{H}$ defined by $K(x, x')$ is the closure of linear span $\{f : f(\cdot) = \sum_{i=1}^r \alpha_i K(x_i, \cdot), \alpha_i \in \mathbb{R}, r \in \mathbb{N}, x_i \in \mathcal{X}\}$ equipped with inner product $\langle f, g \rangle_{\mathcal{H}} = \sum_{ij} \alpha_i \beta_j K(x_i, x_j)$ when $g = \sum_j \beta_j K_{x_j}$. We assume that given the scalar-valued target model $f^* \in \mathcal{H}$, one can uniquely identify a teaching example by its $x^\dagger$ for brevity, $(x^\dagger, y^\dagger) = (x^\dagger, f^*(x^\dagger))$. Let $\mathcal{H}^d = \mathcal{H}_i \times \cdots \times \mathcal{H}_d$ be a RKHS of vector-valued functions $\boldsymbol{f} = [f_i]^d$ with $f_i \in \mathcal{H}$[4], equipped with inner product $\langle \boldsymbol{f}, \boldsymbol{g} \rangle_{\mathcal{H}^d} = \sum_{i=1}^d \langle f_i, g_i \rangle_{\mathcal{H}}$. For simplicity, we use the vector-input $K(\boldsymbol{x}, \boldsymbol{x}') = [K(x_i, x_i')]^d$ to denote kernels in vector-valued RKHS $\mathcal{H}^d$. For a functional $F : \mathcal{H}^d \mapsto \mathbb{R}$, its Fréchet derivative [16, 34, 57] is defined as following:

---

[3]To avoid clutter in the notation, we assume that all learners share same input and output spaces, *i.e.*, $\mathcal{X}_i = \mathcal{X}$ and $\mathcal{Y}_i = \mathcal{Y}$. The results for different input and output spaces can be derived by plugging into specific $\mathcal{X}_i$ and $\mathcal{Y}_i$ directly.

[4]To simplify the notation, we assume that the RKHS of target models are the same for all learners, *i.e.*, $\mathcal{H}_i = \mathcal{H}$.

**Definition 1.** *(Fréchet derivative in vector-valued RKHS) For a vector-valued functional $F$ : $\mathcal{H}^d \mapsto \mathbb{R}$, its Fréchet derivative $\nabla_{\boldsymbol{f}} F[\boldsymbol{f}]$ at $\boldsymbol{f} \in \mathcal{H}^d$ is defined implicitly as $F[\boldsymbol{f} + \epsilon \boldsymbol{g}] = F[\boldsymbol{f}] + \epsilon \langle \nabla_{\boldsymbol{f}} F[\boldsymbol{f}], \boldsymbol{g} \rangle_{\mathcal{H}^d} + \mathcal{O}(\epsilon^2)$ for any $\boldsymbol{g} \in \mathcal{H}^d$ and $\epsilon \in \mathbb{R}$, which is a function in $\mathcal{H}^d$.*

Using the Riesz–Fréchet representation theorem [31, 55], the evaluation functional of vector-valued functions is defined in the following:

**Definition 2.** *For a vector-valued reproducing kernel Hilbert space $\mathcal{H}^d$ with a positive definite kernel $K_{\boldsymbol{x}} \in \mathcal{H}^d$, where $\boldsymbol{x} = [x_{i,j_i}]^d \in \mathcal{X}^d$ and the example index $j_i \in \mathbb{N}_k$, we define evaluation functional $E_{\boldsymbol{x}}[\cdot] : \mathcal{H}^d \mapsto \mathbb{R}$ as*

$$E_{\boldsymbol{x}}[\boldsymbol{f}] = \langle \boldsymbol{f}, K_{\boldsymbol{x}}(\cdot) \rangle_{\mathcal{H}^d} = \sum_{i=1}^d \langle f_i, K_{x_{i,j_i}} \rangle_{\mathcal{H}} = \sum_{i=1}^d f_i(x_{i,j_i}), \boldsymbol{f} = (f_1, \cdots, f_d)^T \in \mathcal{H}^d. \quad (1)$$

**Single-learner nonparametric teaching**. [73] formulates the single-learner nonparametric teaching as a functional minimization over single-learner $\mathbb{D}$ in scalar-valued RKHS:

$$\mathcal{D}^* = \underset{\mathcal{D} \in \mathbb{D}}{\arg\min} \quad \mathcal{M}(\hat{f}^*, f^*) + \lambda \cdot \mathrm{len}(\mathcal{D}) \qquad \text{s.t.} \quad \hat{f}^* = \mathcal{A}(\mathcal{D}), \quad (2)$$

where $\mathcal{M}$ is a disagreement between $\hat{f}^*$ and $f^*$ (*e.g.*, $L_2$ distance defined in RKHS $\mathcal{M}(\hat{f}^*, f^*) = \|\hat{f}^* - f^*\|_{\mathcal{H}}$), $\mathrm{len}(\cdot)$ is the length of the teaching sequence $\mathcal{D}$ (*i.e.*, the ITD defined in [36]) controlled by a regularized constant $\lambda$, and $\mathcal{A}$ denotes the learning algorithm of learners. Usually, $\mathcal{A}(\mathcal{D})$ is taken as empirical risk minimization:

$$\hat{f}^* = \underset{f \in \mathcal{H}}{\arg\min} \, \mathbb{E}_{(x,y) \sim \mathbb{Q}(x,y)} \left[ \mathcal{L}(f(x), y) \right], \quad (3)$$

with single-learner convex loss function $\mathcal{L}$. As introduced in Section 1, iterative teaching [36, 37] focuses on some specific optimization algorithm that the learner adopts [38]. In the nonparametric setting, we consider the functional gradient descent:

$$f^{t+1} \leftarrow f^t - \eta^t \mathcal{G}(\mathcal{L}; f^t; \mathcal{D}^t), \quad (4)$$

where $t = 0, 1, \ldots, T$ serves as an iteration index, $\eta^t > 0$ (*i.e.*, a small constant) denotes the learning rate for the $t$-th iteration, and $\mathcal{G}$ represents the gradient functional evaluated at $\mathcal{D}^t$.

Specifically, [73] investigates the teaching algorithms within a practical teaching protocol and graybox setting. This involves a teacher that has no knowledge about the learner, including the learning rate and specific loss function, but still is able to keep track of the learnt model during each iteration. Two functional teaching algorithms are proposed: Random Functional Teaching (RFT) and Greedy FT (GFT). The former essentially adopts random sampling, and it serves as a simple baseline, which can also be viewed as a functional analogue of stochastic gradient descent [54, 25]. In contrast, GFT picks examples by maximizing the corresponding disagreement between the target and current models [5, 18], and has been shown to be more effective than RFT both theoretically and experimentally.

## 4 MINT: Multi-learner nonparametric teaching

In this section, we begin by defining multi-learner nonparametric teaching as a functional minimization in a vector-valued RKHS. Next, we analyze a vanilla MINT where multiple learners independently and simultaneously learns corresponding components of a vector-valued target function. Lastly, we investigate a communicated MINT where the teacher does not only provide examples but also guide multiple learners in the process of linearly combining present learnt functions.

### 4.1 Teaching settings

To define MINT, we expand scalar-valued target models in single-learner teaching to vector-valued ones and modify other notations to suit the multi-learner setting. More specifically, we redefine functional minimization of Eq. 2 as follows:

$$\boldsymbol{\mathcal{D}}^* = \underset{\boldsymbol{\mathcal{D}} \in \mathbb{D}^d}{\arg\min} \quad \mathcal{M}(\hat{\boldsymbol{f}}^*, \boldsymbol{f}^*) + \lambda \cdot \mathrm{len}(\boldsymbol{\mathcal{D}}) \qquad \text{s.t.} \quad \hat{\boldsymbol{f}}^* = \mathcal{A}(\boldsymbol{\mathcal{D}}), \quad (5)$$

where $\boldsymbol{f}^* \in \mathcal{H}^d$ refers to a vector-valued target model, and other notations bear the same meaning as in Eq. 2. The learning algorithm $\mathcal{A}$ arrives at the following solution:

$$\hat{\boldsymbol{f}}^* = \underset{\boldsymbol{f} \in \mathcal{H}^d}{\arg\min} \, \mathbb{E}_{(\boldsymbol{x}, \boldsymbol{y})} \left[ \mathcal{L}(\boldsymbol{f}(\boldsymbol{x}), \boldsymbol{y}) \right], \tag{6}$$

where $(\boldsymbol{x}, \boldsymbol{y}) \in \mathcal{X}^d \times \mathcal{Y}^d$ and $(\boldsymbol{x}, \boldsymbol{y}) \sim [\mathbb{Q}_i(x_i, y_i)]^d$. Evaluated at an example vector $(\boldsymbol{x}, \boldsymbol{y}) = [(x_{i,j_i}, y_{i,j_i})]^d$ with the example index $j_i \in \mathbb{N}_k$, the multi-learner convex loss $\mathcal{L}$ therein is

$$\mathcal{L}(\boldsymbol{f}(\boldsymbol{x}), \boldsymbol{y}) = \sum_{i=1}^{d} \mathcal{L}_i(f_i(x_{i,j_i}), y_{i,j_i}) = E_{\boldsymbol{x}} \left[ [\mathcal{L}_i(f_i, y_{i,j_i})]^d \right], \tag{7}$$

where $\mathcal{L}_i$ is the convex loss for $i$-th learner. We can also express it as $\langle [1]^d, [\mathcal{L}_i(f_i(x_{i,j_i}), y_{i,j_i})]^d \rangle_{\mathcal{H}^d}$, where the vector $[1]^d$ can be replaced by a weight vector $[w_i]^d \in \mathbb{R}^d$ to adjust the significance of each learner relative to others. Under iterative setting [36, 37] which explores teaching algorithms from the viewpoint of optimization and approximation, we present vector-valued functional gradient descent:

$$\boldsymbol{f}_{A^t}^{t+1} \leftarrow A^t \cdot \boldsymbol{f}^t - \boldsymbol{\eta}^t \odot \boldsymbol{\mathcal{G}}(\mathcal{L}; A^t \cdot \boldsymbol{f}^t; \boldsymbol{\mathcal{D}}^t), \tag{8}$$

where $\odot$ denotes the element-wise multiplication, $\boldsymbol{\eta}^t = [\eta_i]^d = (\eta_1^t, \cdots, \eta_d^t)^T$ is a vector of learning rates that corresponds to $d$ learners and the communication matrix $A^t = \arg\min_{A \in \mathbb{R}^{d \times d}} \|A\boldsymbol{f}^t - \boldsymbol{f}^*\|_{\mathcal{H}^d}$ signifies a matrix with row sums that are equal to one in order to maintain the output's scale. Equivalently, by denoting that $A_{(i,\cdot)}^t = \arg\min_{M_{(i,\cdot)}^t \in \mathbb{R}^{1 \times d}} \|M_{(i,\cdot)}^t \cdot \boldsymbol{f}^t - f_i^*\|_{\mathcal{H}}$, it can also be expressed in a learner-specific (*i.e.*, component-wise) fashion as $f_i^{t+1} \leftarrow A_{(i,\cdot)}^t \cdot f_i^t - \eta_i^t \mathcal{G}_i(\mathcal{L}; A_{(i,\cdot)}^t \cdot \boldsymbol{f}^t; \boldsymbol{\mathcal{D}}^t)$, where $i \in \mathbb{N}_d$ is the learner index.

We investigate MINT in the gray-box setting, which is equivalent to the one considered in [73]. To facilitate the theoretical analysis, we adopt some moderate assumptions regarding $\mathcal{L}_i$ and kernels, which align with those made in [73].

**Assumption 3.** *Each loss $\mathcal{L}_i(f_i), i \in \mathbb{N}_d$ is $L_{\mathcal{L}_i}$-Lipschitz smooth,* i.e., *$\forall f_i, f_i' \in \mathcal{H}$, $x_i \in \mathcal{X}$ and $i \in \mathbb{N}_d$*

$$|E_{x_i} [\nabla_f \mathcal{L}_i(f_i)] - E_{x_i} [\nabla_f \mathcal{L}_i(f_i')]| \le L_{\mathcal{L}_i} |E_{x_i} [f_i] - E_{x_i} [f_i']|,$$

*where $L_{\mathcal{L}_i} \ge 0$ is a constant. To simplify the notation, we assume that $L_{\mathcal{L}_i} = L_{\mathcal{L}}$ for all $i \in \mathbb{N}_d$.*

**Assumption 4.** *Each kernel $K(x, x') \in \mathcal{H}$ is bounded,* i.e., *$\forall x, x' \in \mathcal{X}$, $K(x, x') \le M_K$, where $M_K \ge 0$ is a constant.*

With regards to diverse knowledge domains, we narrow the scope of investigation in this study to the synthesis-based teacher setting [36]. Furthermore, it's worth noting that by limiting the knowledge domain to a specific pool, it can result in multiple learners converging to a suboptimal $\boldsymbol{f}^{*\prime}$, such findings of pool-based teachers are comparable and can be deduced accordingly, as discussed in Remark 7 of [73].

### 4.2 Vanilla multi-learner teaching

In tackling MINT, we begin by examining a basic scenario in which multiple learners concurrently learns corresponding components of a vector-valued target function without communication between them [28, 12], that is, $A^t$ in Eq. 8 is assigned the identity matrix $I_d$. This simplifies Eq. 8 to

$$\boldsymbol{f}^{t+1} \leftarrow \boldsymbol{f}^t - \boldsymbol{\eta}^t \odot \boldsymbol{\mathcal{G}}(\mathcal{L}; \boldsymbol{f}^t; \boldsymbol{\mathcal{D}}^t). \tag{9}$$

In this vanilla setting, multiple learners do not linearly combine learned functions of all learners; rather, it updates its functions by Eq. 9 alone.

In light of the definition of Fréchet derivative in vector-valued RKHS (as presented in Defi.1), we present Chain Rule for vector-valued functional gradients [24] as a Lemma.

**Lemma 5.** *(Chain rule for vector-valued functional gradients) For differentiable functions $G : \mathbb{R} \mapsto \mathbb{R}$ that are functions of functionals $F$, $G(F[\boldsymbol{f}])$, the expression*

$$\nabla_f G(F[\boldsymbol{f}]) = \frac{\partial G(F[\boldsymbol{f}])}{\partial F[\boldsymbol{f}]} \cdot \nabla_{\boldsymbol{f}} F[\boldsymbol{f}] \tag{10}$$

*is usually referred to as the chain rule.*

To obtain the derivative of the evaluation functional [16], we introduce Lemma 6, with the proof of this lemma deferred to Appendix B.

**Lemma 6.** *For an evaluation functional in vector-valued RKHS $E_{\boldsymbol{x}}[\boldsymbol{f}] = \sum_{i=1}^{d} f_i(x_{i,j_i}) : \mathcal{H}^d \mapsto \mathbb{R}$ where $\boldsymbol{x} = [x_{i,j_i}]^d \in \mathcal{X}^d$, its gradient is a $d$-dimensional vector $\nabla_{\boldsymbol{f}} E_{\boldsymbol{x}}[\boldsymbol{f}] = K_{\boldsymbol{x}} = [K_{x_{i,j_i}}]^d \in \mathcal{H}^d$.*

Using Lemma 5 and Lemma 6, we offer an expansion viewpoint on the vector-valued functional gradients of $\mathcal{L}$ [41, 16]: Suppose we have a specific example vector $(\boldsymbol{x}, \boldsymbol{y}) = [(x_{i,j_i}, y_{i,j_i})]^d \in \mathcal{X}^d \times \mathcal{Y}^d$, the gradient $\mathcal{G}$ of the multi-learner loss function $\mathcal{L}$ w.r.t. the vector-valued model $\boldsymbol{f}$ can be expressed by

$$\mathcal{G}(\mathcal{L}; \boldsymbol{f}; (\boldsymbol{x}, \boldsymbol{y})) = \left[ \partial \mathcal{L}_i / \partial f_i |_{f_i(x_{i,j_i}), y_{i,j_i}} K_{x_{i,j_i}} \right]^d. \tag{11}$$

We also broaden the applicability of RFT and GFT from their single-learner versions [73] to a multi-learner one. Under this context, RFT involves randomly picking examples for each learner, while GFT selects examples that satisfy

$$\left( \boldsymbol{x}^{t^*} = \underset{[x_i]^d \in \mathcal{X}^d}{\arg\max} \left\| \left[ \partial \mathcal{L}_i / \partial f_i |_{f_i^t(x_i)} \right]^d \right\|_{\mathcal{H}^d}, \quad \boldsymbol{y}^{t^*} = [y^{t^*}]^d = \left[ f_i^* \left( x_i^{t^*} \right) \right]^d \right) \tag{12}$$

To avoid clutter in the notation, our examination is restricted to the selection of a single example for each learner during every iteration, and we provide the pseudo code in Appendix A.

In the upcoming discussion, we shall present our theoretical examination of the convergence performance of multi-learner RFT and GFT. Our approach differs from [73] as we focus on RFT's average performance by introducing the expectation operation over random sampling. This helps us gain valuable insights by averaging out the impact of randomness. Recall the teaching settings (Eq. 6, 9), we then proceed with our analysis of RFT's per-iteration reduction concerning $\mathcal{L}$.

**Lemma 7.** *(Sufficient Descent for multi-learner RFT) Suppose there are $d$ learners, and the example mean for each learner is $\mu_i = \mathbb{E}_{x_i \sim \mathbb{P}_i(x_i)}(x_i) < \infty$, and the variance $\sigma_i^2 = \mathbb{E}_{x_i \sim \mathbb{P}_i(x_i)}(x_i - \mu_i)^2 < \infty, i \in \mathbb{N}_d$. Under the assumptions outlined in both 3 and 4, if $\eta_i^t \leq \frac{1}{2L_{\mathcal{L}} \cdot M_K}$ for all $i \in \mathbb{N}_d$, then RFT teachers can, on average, reduce the multi-learner loss $\mathcal{L}(\boldsymbol{f})$ by:*

$$\mathbb{E}_{\boldsymbol{x} \sim [\mathbb{P}_i(x_i)]^d} \left[ \mathcal{L}(\boldsymbol{f}^{t+1}) - \mathcal{L}(\boldsymbol{f}^t) \right] \leq -\frac{\tilde{\eta}^t}{2} \sum_{i=1}^{d} (m_{i,t}(\mu_i) + \frac{m_{i,t}''(\mu_i)}{2} \sigma_i^2), \tag{13}$$

*where $\tilde{\eta}^t = \min_{i \in \mathbb{N}_d} \eta_i^t$ and $m_{i,t}(\dot{x}) := E_{\dot{x}}[(\nabla_f \mathcal{L}_i(f)|_{f=f_i^t})^2]$.*

Intuitively, $m_{i,t}(\mu_i)$ serves as a measure of the gradient's magnitude (loss $\mathcal{L}_i$ w.r.t. $f_i^t$) for the $i$-th learner at the example mean $\mu_i$ in the $t$-th iteration, and as $f_i^t$ approaches $f_i^*$, $m_{i,t}(\mu_i)$ becomes increasingly small. According to Lemma 7, as proven in Appendix B, the average reduction of multi-learner loss $\mathcal{L}$ per iteration is constrained by a negative upper bound. To be more precise, this upper bound is determined by a range of elements, such as the learning rate, the count of learners, the example variance $\sigma_i^2$ and the gradient of $\mathcal{L}_i$ at the example mean $\mu_i$ ($i \in \mathbb{N}_d$), and these elements are independent of each another. When the gradient at each $\mu_i$ is large, RFT on average can reduce $\mathcal{L}$ by a significant amount. Meanwhile, the variance also has an impact on the reduction and this is governed by a diminutive constant of $m_{i,t}''(\mu_i)/2$.

The least helpful teaching scenario in the $t$-th iteration across the $d$ learners is represented by $\min_{i \in \mathbb{N}_d} \left( m_{i,t}(\mu_i) + m_{i,t}''(\mu_i)\sigma_i^2/2 \right)$, which has the smallest gradient at the example mean. It is observed from Lemma 7 that the reduction of $\mathcal{L}$ per iteration is, at a minimum, $d$ times greater than that of the worst-case scenario $-\tilde{\eta}^t/2 \cdot \min_{i \in \mathbb{N}_d} \left( m_{i,t}(\mu_i) + m_{i,t}''(\mu_i)\sigma_i^2/2 \right)$. In other words, when multi-learner RFT achieves a stationary point in the worst-case scenario, the multi-learner loss $\mathcal{L}$ reaches convergence as well. This indicates that the convergence rate of multi-learner RFT is at least as fast as in the single-learner worst-case scenario (also faster than repeatedly teaching).

Introducing the expectation operation enables us to eliminate the randomness that arises from random sampling. In contrast to [73], which determines the decrease based on the discrepancy at specific but randomly chosen example $x^t$ (involving randomness), we establish that the decrease on average is determined by the mean and variance. This valuable insight is important to understand the fundamentals of RFT, which is not considered in [73].

**Theorem 8.** *(Convergence for multi-learner RFT) Suppose the vector-valued model for multiple learners is initialized with $\boldsymbol{f}^0 \in \mathcal{H}^d$ and returns $\boldsymbol{f}^t \in \mathcal{H}^d$ after $t$ iterations, we have the upper bound of $\min_{i \in \mathbb{N}_d} \left( m_{i,t}(\mu_i) + m''_{i,t}(\mu_i)\sigma_i^2/2 \right)$ w.r.t. $t$:*

$$\min_{i \in \mathbb{N}_d} \left( m_{i,t-1}(\mu_i) + m''_{i,t-1}(\mu_i)\sigma_i^2/2 \right) \leq 2\mathbb{E}_{\boldsymbol{x} \sim [\mathbb{P}_i(x_i)]^d} \left[ \mathcal{L}(\boldsymbol{f}^0) \right] / (d\dot{\eta}t), \tag{14}$$

*where $0 < \dot{\eta} = \min_{l \in \{0\} \bigcup \mathbb{N}_{t-1}} \tilde{\eta}^l \leq 1/(2L_\mathcal{L} \cdot M_K)$, and given a small constant $\epsilon > 0$ it would take approximately $\mathcal{O}(2\mathbb{E}_{\boldsymbol{x} \sim [\mathbb{P}_i(x_i)]^d} \left[ \mathcal{L}\left(\boldsymbol{f}^0\right) \right] / (d\dot{\eta}\epsilon))$ iterations to reach a stationary point.*

The proof for Theorem 8 can be found in Appendix B. Theorem 8 tells that the minimum of the non-negative term within the upper bound in Theorem 7, which is $\min_{i \in \mathbb{N}_d} \left( m_{i,t}(\mu_i) + m''_{i,t}(\mu_i)\sigma_i^2/2 \right)$, is also upper bounded, and the iterative teaching dimension is $2\mathbb{E}_{\boldsymbol{x} \sim [\mathbb{P}_i(x_i)]^d} \left[ \mathcal{L}(\boldsymbol{f}^0) \right] / (d\dot{\eta}\epsilon)$.

In comparison to RFT, GFT achieves a larger reduction in multi-learner loss $\mathcal{L}$ per iteration, suggesting a faster convergence rate and a lesser number of iterations required to achieve convergence.

**Lemma 9.** *(Sufficient Descent for multi-learner GFT) Under Assumption 3 and 4, if $\eta_i^t \leq \frac{1}{2L_\mathcal{L} \cdot M_K}$ for all $i \in \mathbb{N}_d$, the GFT teachers can achieve a greater reduction in the multi-learner loss $\mathcal{L}$:*

$$\mathbb{E}_{\boldsymbol{x} \sim [\mathbb{P}_i(x_i)]^d} \left[ \mathcal{L}(\boldsymbol{f}^{t+1}) - \mathcal{L}(\boldsymbol{f}^t) \right] \leq -\frac{\tilde{\eta}^t}{2} \sum_{i=1}^d m_{i,t}(x_i^{t*}), \tag{15}$$

*where $\tilde{\eta}^t$ and $m_{i,t}(\cdot)$ retain their previous meaning.*

The proof of the Lemma 9 is presented in Appendix B. GFT selects examples with the steepest gradient, which leads to $m_{i,t}(x_i^{t*}) \geq \left( m_{i,t}(\mu_i) + m''_{i,t}(\mu_i)\sigma_i^2/2 \right)$ for each learner. Consequently, it can be observed that per-iteration $\mathcal{L}$ reduction of GFT has a tighter bound compared to RFT. This is due to the fact that GFT uses a greedy approach to select examples that maximizes the norm of difference between the current and target models. This allows the learners to take a larger step forward $\boldsymbol{f}^*$ in per iteration. The tighter bound provides theoretical evidence supporting the effectiveness of GFT, which is consistent with the findings in the single-learner teaching [73].

**Theorem 10.** *(Convergence for multi-learner GFT) Suppose the vector-valued model for multiple learners is initialized with $\boldsymbol{f}^0 \in \mathcal{H}^d$ and returns $\boldsymbol{f}^t \in \mathcal{H}^d$ after $t$ iterations, we have the upper bound of $\min_{i \in \mathbb{N}_d} m_{i,t}(x_i^{t*})$ w.r.t. $t$:*

$$\min_{i \in \mathbb{N}_d} m_{i,t-1}(x_i^{t-1*}) \leq \frac{2}{d\dot{\eta}t}\mathbb{E}_{\boldsymbol{x} \sim [\mathbb{P}_i(x_i)]^d} \left[ \mathcal{L}(\boldsymbol{f}^0) \right] + \frac{1}{dt} \sum_{l=0}^{t-1} \sum_{i=1}^d \left( \|x_i^{l*} - \mu_i\|_2 \right), \tag{16}$$

*where $\dot{\eta}$ has the same definition as before.*

It follows from Lemma 7 and 9 that when $x_i^{t*}$ is close to $\mu_i$ for $i \in \mathbb{N}_d$, then GFT and RFT perform similarly. In Theorem 10 (The proof is given in Appendix B), we theoretically show this relation by introducing the distance between $x_i^{t*}$ and $\mu_i$, which provides a deep insight of the difference between RFT and GFT that is not considered in [73]. Specifically, the per-iteration loss reduction under both RFT and GFT has negative upper bounds, and the difference between these two upper bounds can be seen by comparing Theorem 8 and Theorem 10. From a qualitative perspective, GFT can achieve better convergence speed-up because its negative upper bound can take smaller values than that of RFT. This gap is characterized by $\frac{1}{dt} \sum_{l=0}^{t-1} \sum_{i=1}^d \left( \|x_i^{l*} - \mu_i\|_2 \right)$ which is the cumulative distance between select $x_i^{t*}$ and mean $\mu_i$ for all learners and averaged over iterations. We emphasize that the purpose of our results is to show the difference between RFT and GFT, rather than proving that GFT always achieves better convergence than RFT (which is not always true). By comparing Theorem 8 and Theorem 10, we can learn that it is possible for GFT to have larger per-iteration loss reduction than RFT. However, we also recognize the intrinsic difficulty to show the exact conditions such that GFT can always be better than RFT. In contrast to our results, the parametric case (*e.g.*, [36]) also has not obtained the necessary and sufficient conditions for greedy teaching to be better than random teaching. More generally, [38] also considers some alternative teaching strategies other than the greedy teaching, such as the parameterized teaching with a multi-iteration reward function. Despite not being able to fully characterize the difference of convergence rate between GFT and RFT, our existing theoretical analysis still poses an important open problem: *when and how can GFT provably achieve faster convergence than RFT?*

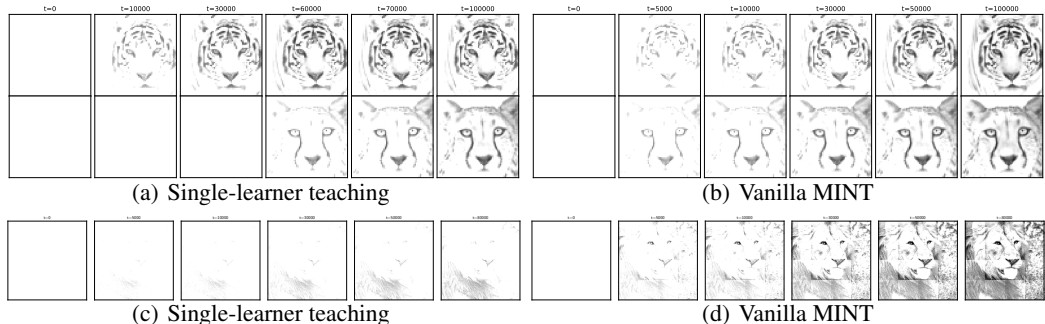

(a) Single-learner teaching            (b) Vanilla MINT

(c) Single-learner teaching            (d) Vanilla MINT

Figure 2: Comparison between single-learner teaching and MINT. (a) Repeatedly invoking single-learner GFT: teaching a white tiger at first and subsequently teaching a cheetah. (b) Simultaneous teaching of a white tiger and a cheetah by GFT. (c) Single-learner teaching of the lion. (d) Partitioning a single lion image into 16 pieces and teaching them concurrently.

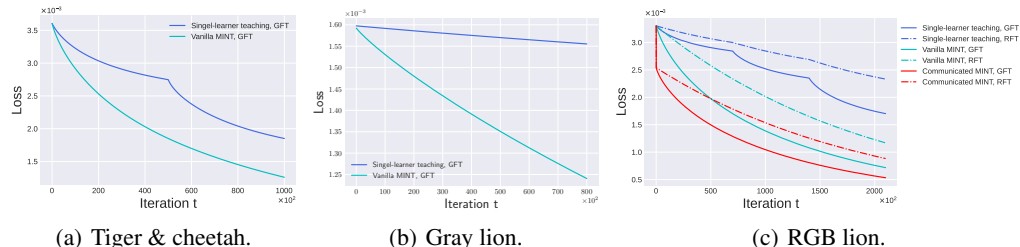

(a) Tiger & cheetah.        (b) Gray lion.        (c) RGB lion.

Figure 3: Comparison of convergence performance between single-learner teaching and MINT. (a) is corresponding to (a)-(b) in Figure 2. (b) is for (c)-(d) in Figure 2. (c) pertains to teaching of a colored lion.

## 4.3 Communicated multi-learner teaching

An infant would often compose previously learnt knowledge in order to grasp a new target concept, such as understanding what a zebra is by combining the learnt ideas of horses and black-and-white stripes. Such an efficient learning motivates us to explore the idea of communicated MINT, which enables the communication between learners. In other words, multiple learners can execute linear combination on the currently learnt functions of all learners [21, 23, 77, 12], that is, $A^t$ is no longer constrained to be an identity matrix.

In practice, to direct this communication, the teacher can utilize a two-layer perceptron (MLP) to derive the matrix $A^t$ in Eq. 8 by searching a matrix $A$ that minimizes $\|A\boldsymbol{f}^t - \boldsymbol{f}^*\|_{\mathcal{H}^d}$ as much as possible, which is an addition step beyond example selection in each iteration.

**Proposition 11.** *If the proximity between $\boldsymbol{f}^t$ and $\boldsymbol{f}^*$ is sufficiently close, meaning that $\|\boldsymbol{f}^t - \boldsymbol{f}^*\|_{\mathcal{H}^d} \leq \epsilon$ where $\epsilon$ is a tiny positive constant, then $A^t$ equals the identity matrix $I_d$.*

The proof of Prop.11 is given in Appendix B. This suggests that there is no need for MLP to be used in solving matrix $A^t$ in every iteration, but only at the beginning, because as the iterations progress, $\boldsymbol{f}^t$ will approach near to $\boldsymbol{f}^*$.

**Lemma 12.** *Under Assumption 3, the communication across learners will result in a reduction of the multi-learner convex loss $\mathcal{L}$ by $0 \leq \mathcal{L}(\boldsymbol{f}^t) - \mathcal{L}(A^t\boldsymbol{f}^t) \leq 2L_{\mathcal{L}}\|\boldsymbol{f}^t - \boldsymbol{f}^*\|_{\mathcal{H}^d}$.*

Proof of Lemma 12 is given in Appendix B. The difference in $\mathcal{L}$ between the case where the communication exists and that where it doesn't is lower bounded by zero and upper bounded by the distance between $\boldsymbol{f}^t$ and $\boldsymbol{f}^*$. This suggests that if $\boldsymbol{f}^t$ is far from $\boldsymbol{f}^*$, then matrix $A^t$ can potentially decrease $\mathcal{L}$ significantly at the best case while not causing any increase at the worst case.

**Theorem 13.** *Suppose the communication in the $t$-th iteration of multiple learners is denoted by the matrix $A^t$ and returns $\boldsymbol{f}_{A^t}^{t+1} \in \mathcal{H}^d$, for both RFT and GFT we have:*

$$\mathbb{E}_{\boldsymbol{x}\sim[\mathbb{P}_i(x_i)]^d}\left[\mathcal{L}(\boldsymbol{f}_{A^t}^{t+1}) - \mathcal{L}(\boldsymbol{f}^t)\right] \leq \mathbb{E}_{\boldsymbol{x}\sim[\mathbb{P}_i(x_i)]^d}\left[\mathcal{L}(\boldsymbol{f}_{A^t}^{t+1}) - \mathcal{L}(A^t\boldsymbol{f}^t)\right] \leq 0. \qquad (17)$$

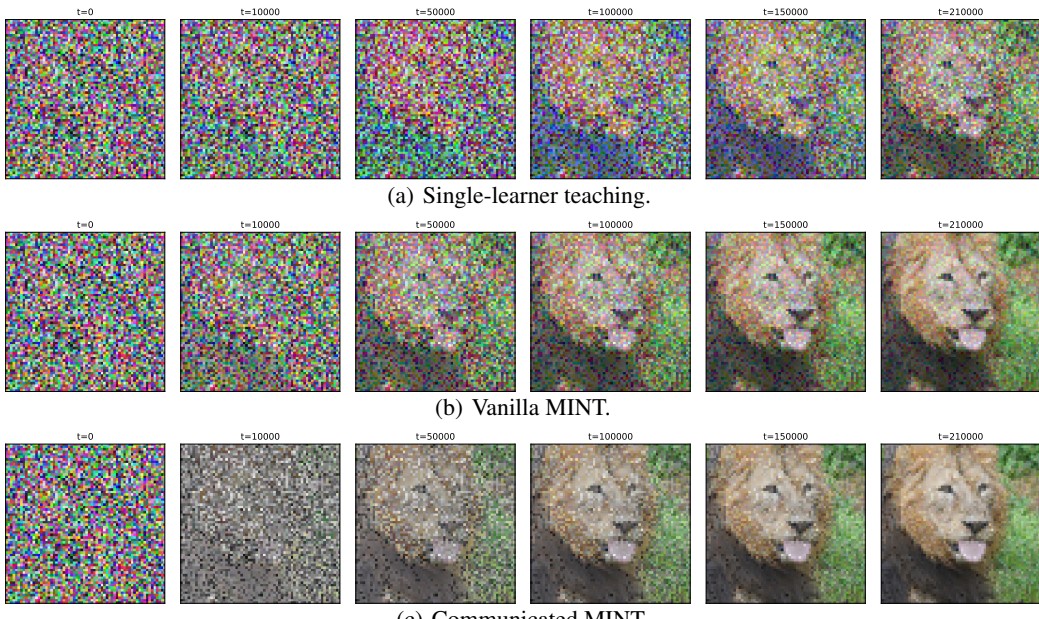

Figure 4: Visualization of $\boldsymbol{f}^t$ taught by GFT. Starting from a random initialization, the communicated multi-learner GFT help multiple learners learn a more clear image than the vanilla one followed by single-learner one.

Proof of Theorem 13 is in Appendix B. This shows that the addition of communication has led to an improvement in model updates, which is evident from the larger loss discrepancy between $\boldsymbol{f}_{A^t}^{t+1}$ and $\boldsymbol{f}^t$ compared to the difference observed between $\boldsymbol{f}_{A^t}^{t+1}$ and $A^t\boldsymbol{f}^t$.

## 5   Experiments and Results

Testing the teaching of a multi-learner (vector-valued) target model, MINT presents more satisfactory performance than repeatedly carrying out the single-learner teaching, which is consistent with our theoretical findings. Detailed configurations and supplementary experiments are given in the Appendix C.

**MINT in gray scale.** A grayscale figure can be viewed as a 3D surface where the $z$ axis corresponds to the level of gray, while the $x, y$ axes depict the placement of pixels [73]. We consider two scenarios: one involves the simultaneous teaching of a tiger and a cheetah figure, while the other focuses on the teaching of a lion. After comparing (a) and (b) in Figure 2, we see that when teaching two target functions by GFT simultaneously, the vanilla MINT requires almost half the number of cost iterations compared to single-learner teaching, which is also evident from the loss plot shown in Figure 3 (a). By comparing (c) and (d) in Figure 2, we can observe that dividing a single-learner target figure into smaller pieces and recasting them into MINT can significantly improve the efficiency, which is also demonstrated by the loss plot in Figure 3 (b).

**MINT in three (RGB) channels.** To further demonstrate the benefits of communication, we examine with a lion image with three channels in RGB format. The loss plot in Figure 3 (c) reveals that the most efficient teaching is the communicated MINT for both RFT and GFT. The vanilla MINT and single-learner teaching follow in order of decreasing efficiency. Furthermore, as anticipated, the multi-learner GFT proves to be more efficient compared to RFT. One intriguing observation is that the communicated MINT leads to a significant reduction in multi-learner loss at the outset, which aligns with our theoretical findings in Lemma 12 and confirms the validity of Prop.11 that $A^t$ could eventually become an identity matrix after numerous iterations. Figure 4 compares the specific learnt $\boldsymbol{f}^t$ for three versions of GFT during each iteration, wherein we observe that MINT consistently outperforms the single-learner one, and the learnt image under the communicated MINT is more clear compared to that of the vanilla one. To be more persuasive, we also offer detailed and additional

---

Our source code is available at https://github.com/chen2hang/MINT_NonparametricTeaching.

experiments in Appendix, including channel-wise visualization of specific $\boldsymbol{f}^t$ (Figure 7), RFT-taught $\boldsymbol{f}^t$ (Figure 8-9) and teaching multiple learners with a particular initialization of $\boldsymbol{f}^0$ (Figure 11-10), which includes an extreme case that only one-time communication is sufficient to help multiple learners learn $\boldsymbol{f}^*$ (Figure 15).

## 6 Concluding Remarks and Future Work

In this paper, we seek to address a practical limitation of current nonparametric iterative machine teaching by enabling the teaching of multi-learner (vector-valued) target models. This generalization of teaching ability involves generalizing the model space from space of scalar-valued functions to that of vector-valued functions. In order to address multi-learner nonparametric teaching, we start by analyzing a vanilla MINT where the teacher picks examples based on a vector-valued target function such that multiple learners can learn its components simultaneously. Additionally, we consider the communicated MINT (*i.e.*, multiple learners are allowed to carry out linear combination on current learnt functions) for further exploration. Through both theoretical analysis and empirical evidence, we demonstrate that the communicated MINT is more efficient than the vanilla MINT.

Moving forward, it could be interesting to explore other practical aspects related to nonparametric teaching. This will involve a deeper theoretical understanding and the development of more efficient teaching algorithms. Besides, it would be intriguing to establish connections between MINT and multi-output neural networks, which can further enhance its practical applications such as knowledge distillation. Moreover, generating teaching examples with a surrogate objective that does not need a target model (*e.g.*, black-box teaching) is also an important direction (*e.g.*, [17, 72]). More generally, (iterative) machine teaching is intrinsically connected to the recent popular data-centric AI. Understanding data-centric learning (*e.g.*, text prompting, data augmentation, data distillation) may require a deeper understanding towards (iterative) machine teaching.

## Acknowledgements

This work was supported in part by National Natural Science Foundation of China (Grant Number: 62206108), in part by Maritime AI Research Programme (SMI-2022-MTP-06) and AI Singapore OTTC Grant (AISG2-TC-2022-006), and in part by the Research Grants Council of the Hong Kong Special Administrative Region (Grant 16200021).

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

# Appendix

## A  Additional Discussions

### Broader Impact

This paper is to address a practical limitation of recently introduced Nonparametric Iterative Machine Teaching from the theoretical aspect, which is to enable multi-learner teaching. We also provide empirical evidence to demonstrate that multi-learner nonparametric teaching is effective in addressing such a limitation. Although we hope that the outcomes of this study will enlighten the theoretical community, we do not expect any immediate effects on society from this work.

### Pseudo code for multi-learner RFT and GFT

The pseudo code for RFT and GFT in the vanilla and communicated MINT is given as following:

---

**Algorithm 1** Greedy (Random) Functional Teaching for the Communicated (Vanilla) MINT

---

**Input:** Target $\boldsymbol{f}^* \in \mathcal{H}^d$, initial $\boldsymbol{f}^0 \in \mathcal{H}^d$, small constants $\epsilon, \epsilon_0 > 0$ and maximal iteration numbers $T, T_0$.

Set $\boldsymbol{f}^t \leftarrow \boldsymbol{f}^0, t = 0$.

**while** $t \leq T$ and $\|\boldsymbol{f}^t - \boldsymbol{f}^*\|_{\mathcal{H}^d} \geq \epsilon$ **do**

    **The teacher** is not only to construct the communication matrix but alos to select the teaching set for multiple learners:

    `// Construction of the communication matrix.`

    Initialize a two-layer perceptron with a linear layer, *i.e.*, initialize the communication matrix with an indentity one ($A = I_d$), and set $t_0 = 0$.

    **while** $t_0 \leq T_0$ and $\|A\boldsymbol{f}^t - \boldsymbol{f}^*\|_{\mathcal{H}^d} \geq \epsilon_0$ **do**

    |   Train the linear weight ($A$) of this two-layer perceptron, such that $\|A\boldsymbol{f}^t - \boldsymbol{f}^*\|_{\mathcal{H}^d}$ decreases.

    **end**

    `/* For the Vanilla MINT, omit the above procedure of solving for the`
    `   communication matrix, just need to set ` $A = I_d$`.                      */`

    `// Selection of the teaching set.`

    Initialize the teaching set $\mathcal{D} = \emptyset$;

    Pick $x_i^{t*} \in \mathcal{X}_i, i \in \mathbb{N}_d$ with the maximal difference between $f_i^t$ and $f_i^*$ for multiple learners simultaneously:
$$x_i^{t*} = \arg\max_{x_i^t \in \mathcal{X}_i} \left| f_i^t(x_i^t) - f_i^*(x_i^t) \right|;$$

    `/* For random functional teaching, Pick ` $x_i^{t*} \in \mathcal{X}_i, i \in \mathbb{N}_d$ ` randomly and`
    `   concurrently.                                                            */`

    Add $\left(x_i^{t*}, y_i^{t*} = f_i^*\left(x_i^{t*}\right)\right)$ into the teaching set $\mathcal{D}$.

    Provide $A$ and $\mathcal{D}$ to multiple learners.

    **The learners** update $\boldsymbol{f}^t$ based on received $A^t = A$ and $\mathcal{D}^t = \mathcal{D}$:

    $\boldsymbol{f}_{A^t}^{t+1} \leftarrow A^t \cdot \boldsymbol{f}^t - \boldsymbol{\eta}^t \odot \mathcal{G}(\mathcal{L}; A^t \cdot \boldsymbol{f}^t; \mathcal{D}^t)$.

    Set $t \leftarrow t + 1$.

**end**

---

### Further discussion about the idea behind MINT

In practical scenarios, it is a commonly accepted fact that a single object can possess multiple characteristics. For instance, when describing a fruit, we often take into account its various attributes such as shape, color, and texture. This highlights the notion that limiting oneself to scalar-vector functions in nonparametric iterative machine teaching would not be enough to capture the complexity of real-world data. We thus extend the current nonparametric teaching from the single-learner version to a multi-learner one (MINT) by considering vector-valued functions. Compared to parameterized

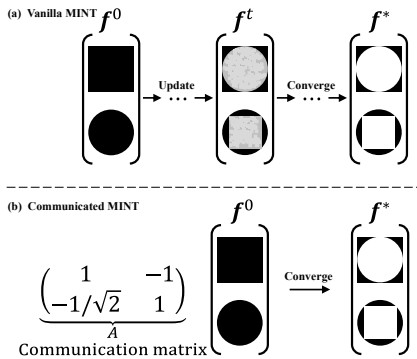

Figure 5: Comparison between vanilla and communicated MINT.

teaching, MINT can be seen as a type of extension which involves replacing the vector unit from values to functions.

The idea of communicated MINT partially comes from linear algebra. If $\boldsymbol{f}^0 \in \mathcal{H}^d$ forms a basis of $\mathcal{H}^d$, and $\boldsymbol{f}^*$ can be linearly expressed by this basis, then it is noteworthy that learners can successfully learn $\boldsymbol{f}^*$ with only one-time communication.

Let us consider a scenario in which the components of $\boldsymbol{f}^0$ are from different function families. Suppose we take $\boldsymbol{f}^0 = (e^x, \sin x, x)^T$ and $\boldsymbol{f}^* = (2e^x - \sin x - x, \sin x + 2x, -e^x + \sin x + x)^T$, it holds that

$$
\begin{pmatrix} 2e^x - \sin x - x \\ \sin x + 2x \\ -e^x + \sin x + x \end{pmatrix} = \underbrace{\begin{pmatrix} 2 & -1 & -1 \\ 0 & 1 & 2 \\ -1 & 1 & 1 \end{pmatrix}}_{\text{communication matrix } A} \cdot \begin{pmatrix} e^x \\ \sin x \\ x \end{pmatrix},
\tag{18}
$$

which indicates that the teacher can teach $\boldsymbol{f}^*$ to multiple learners through providing the communication matrix $A$ within one iteration. To draw a practical analogy, we can view $e^{kx}$ as denoting shape, $\sin kx$ as indicating color, and $x^{1/k}$ as representing texture.

Another scenario is when the components of $\boldsymbol{f}^0$ belong to the same function family. Consider the scenario where two learners are familiar with the shapes of a circle and a square, respectively, and they need to learn two different shapes as shown in Figure 5. In this case, we can model it by assuming that $f_j^0, j \in \mathbb{N}_2$ belong to the *cosine* family, *i.e.*, $\boldsymbol{f}^0 = (\sqrt{2}\cos(2\pi x), \sqrt{2}\cos(4\pi x))^T$. Here, $f_j^0$ are not linearly related, which is a property of basis. The target function is denoted as $\boldsymbol{f}^* = (\sqrt{2}\cos(2\pi x) - \sqrt{2}\cos(4\pi x), -\cos(2\pi x) + \sqrt{2}\cos(4\pi x))^T$. It is trivial to derive that

$$
\begin{pmatrix} \sqrt{2}\cos(2\pi x) - \sqrt{2}\cos(4\pi x) \\ -\cos(2\pi x) + \sqrt{2}\cos(4\pi x) \end{pmatrix} = \begin{pmatrix} 1 & -1 \\ -1/\sqrt{2} & 1 \end{pmatrix} \cdot \begin{pmatrix} \sqrt{2}\cos(2\pi x) \\ \sqrt{2}\cos(4\pi x) \end{pmatrix}.
\tag{19}
$$

By referring to the illustration in Figure 5, we can compare the performance of vanilla MINT and communicated MINT. It can be observed that the communicated MINT is capable of helping to learn $\boldsymbol{f}^*$ within just one iteration, whereas the vanilla MINT requires numerous iterations to achieve the same.

## B   Detailed Proofs

Our suggestion for further reading on functional calculus would be to consult the literature [24, 16] and the references therein.

**Proof of Lemma 6**   We firstly define a function $\boldsymbol{q}$ by adding a small disturbance $\epsilon\boldsymbol{g}$ ($\epsilon \in \mathbb{R}, \boldsymbol{g} \in \mathcal{H}^d$) to $\boldsymbol{f} \in \mathcal{H}^d$, that is, $\boldsymbol{q} = \boldsymbol{f} + \epsilon\boldsymbol{g}$. We see that $\boldsymbol{q} \in \mathcal{H}^d$ since vector-valued RKHS is closed under addition and scalar multiplication. For a evaluation functional $E_{\boldsymbol{x}}[\boldsymbol{f}] = \sum_{i=1}^d f_i(x_{i,j_i}) : \mathcal{H}^d \mapsto \mathbb{R}$ where $\boldsymbol{x} = [x_{i,j_i}]^d \in \mathcal{X}^d$, we thus can evaluate $\boldsymbol{q}$ at $\boldsymbol{x}$ as

$$
\begin{aligned}
E_{\boldsymbol{x}}[\boldsymbol{q}] &= E_{\boldsymbol{x}}[\boldsymbol{f} + \epsilon\boldsymbol{g}] \\
&= E_{\boldsymbol{x}}[\boldsymbol{f}] + \epsilon E_{\boldsymbol{x}}[\boldsymbol{g}] + 0 \\
&= E_{\boldsymbol{x}}[\boldsymbol{f}] + \epsilon\langle K(\boldsymbol{x}, \cdot), \boldsymbol{g} \rangle_{\mathcal{H}^d} + 0
\end{aligned}
\tag{20}
$$

Recall the definition of the Fréchet derivative in a vector-valued reproducing kernel Hilbert space (RKHS), which is defined implicitly and can be found in Definition 1, $E_{\boldsymbol{x}}[\boldsymbol{f} + \epsilon\boldsymbol{g}] = E_{\boldsymbol{x}}[\boldsymbol{f}] + \epsilon\langle\nabla_{\boldsymbol{f}} E_{\boldsymbol{x}}[\boldsymbol{f}], \boldsymbol{g}\rangle_{\mathcal{H}^d} + \mathcal{O}(\epsilon^2)$, it follows from Eq. 20 that the gradient of a evaluation functional in vector-valued RKHS $\mathcal{H}^d$ is $\nabla_{\boldsymbol{f}} E_{\boldsymbol{x}}[\boldsymbol{f}] = K_{\boldsymbol{x}} = [K_{x_{i,j_i}}]^d$.

∎

**Proof of Lemma 7**    Note that one example $(x, y) \sim \mathbb{Q}_i(x, y)$ can be uniquely identified by its $x$, we take expectation on $x \sim \mathbb{P}(x)$ only for the sake of simplicity. It follows from the convexity of $\mathcal{L}_i$ and the definition of Fréchet derivative in Definition 1 that we have

$$
\begin{aligned}
&\mathbb{E}_{x_i \sim \mathbb{P}_i(x_i)}\left[\mathcal{L}_i(f_i^{t+1}) - \mathcal{L}_i(f_i^t)\right] \\
\leq\ &\mathbb{E}_{x_i \sim \mathbb{P}_i(x_i)}\left[\langle f_i^{t+1} - f_i^t, \nabla_f \mathcal{L}_i(f)|_{f=f_i^{t+1}}\rangle_{\mathcal{H}}\right] \\
=\ &\mathbb{E}_{x_i \sim \mathbb{P}_i(x_i)}\left[-\eta_i^t E_{x_i^t}\left[\nabla_f \mathcal{L}_i(f)|_{f=f_i^t}\right] \cdot E_{x_i^t}\left[\nabla_f \mathcal{L}_i(f)|_{f=f_i^{t+1}}\right]\right] \\
=\ &-\eta_i^t \cdot \mathbb{E}_{x_i \sim \mathbb{P}_i(x_i)}\left[E_{x_i^t}\left[\nabla_f \mathcal{L}_i(f)|_{f=f_i^t} \cdot \nabla_f \mathcal{L}_i(f)|_{f=f_i^{t+1}}\right]\right].
\end{aligned}
\tag{21}
$$

Under $L_\mathcal{L}$-Lipschitz smooth Assumption 3 and bounded kernel function Assumption 4, we can show that

$$
E_{x_i^t}\left[\nabla_f \mathcal{L}_i(f)|_{f=f_i^t} \cdot \nabla_f \mathcal{L}_i(f)|_{f=f_i^{t+1}}\right] \geq \left(3/4 - L_\mathcal{L}^2(\eta_i^t)^2(M_K)^2\right) E_{x_i^t}\left[\left(\nabla_f \mathcal{L}_i(f)|_{f=f_i^t}\right)^2\right].
\tag{22}
$$

For succinctness, we define

$$
\left(\nabla_f \mathcal{L}_i(f)|_{f=f_i^t}\right)^2 := \nabla_f^2 \mathcal{L}_i(f)|_{f=f_i^t}
\tag{23}
$$

and

$$
m_{i,t}(\dot{x}) := E_{\dot{x}}\left[\nabla_f^2 \mathcal{L}_i(f)|_{f=f_i^t}\right] = E_{\dot{x}}\left[\left(\nabla_f \mathcal{L}_i(f)|_{f=f_i^t}\right)^2\right].
\tag{24}
$$

Then, we can apply Taylor expansion for $m_{i,t}(x_i^t)$ on $\mu_i = \mathbb{E}_{x_i \sim \mathbb{P}_i(x_i)}(x_i^t)$ and derives

$$
m_{i,t}(x_i^t) = m_{i,t}(\mu_i) + m_{i,t}'(x_i^t)(x_i^t - \mu_i) + \frac{m_{i,t}''(\mu_i)}{2}(x_i^t - \mu_i)^2 + R_2(x_i^t),
\tag{25}
$$

where the remainder $R_2(x_i^t)$ usually is omitted, and we assume $m_{i,t}(x_i^t)$ is 2-times differentiable. We see that evaluated at mean $\mu_i$, $m_{i,t}(\mu_i)$ is monotonically decreasing w.r.t. iteration $t$. Combining Eq. 21, 22 and 25, we have

$$
\begin{aligned}
&\mathbb{E}_{x_i \sim \mathbb{P}_i(x_i)}\left[\mathcal{L}_i(f_i^{t+1}) - \mathcal{L}_i(f_i^t)\right] \\
\leq\ &-\eta_i^t \cdot \mathbb{E}_{x_i \sim \mathbb{P}_i(x_i)}\left[\left(3/4 - L_\mathcal{L}^2(\eta_i^t)^2(M_K)^2\right) E_{x_i^t}\left[\nabla_f^2 \mathcal{L}_i(f)|_{f=f_i^t}\right]\right] \\
=\ &-\eta_i^t \left(3/4 - L_\mathcal{L}^2(\eta_i^t)^2(M_K)^2\right) \cdot \left(m_{i,t}(u_i) + \frac{m_{i,t}''(\mu_i)}{2}\sigma_i^2\right),
\end{aligned}
\tag{26}
$$

where $\sigma_i^2 = \mathbb{E}_{x_i \sim \mathbb{P}_i(x_i)}(x_i^t - \mu_i)^2$ is the variance of $x_i$. Therefore,

$$
\begin{aligned}
&\mathbb{E}_{\boldsymbol{x} \sim [\mathbb{P}_i(x_i)]^d}\left[\mathcal{L}(\boldsymbol{f}^{t+1}) - \mathcal{L}(\boldsymbol{f}^t)\right] \\
=\ &\mathbb{E}_{\boldsymbol{x} \sim [\mathbb{P}_i(x_i)]^d}\left[\sum_{i=1}^d \mathcal{L}_i(f_i^{t+1}) - \sum_{i=1}^d \mathcal{L}_i(f_i^t)\right] \\
=\ &\sum_{i=1}^d \mathbb{E}_{x_i \sim \mathbb{P}_i(x_i)}\left[\mathcal{L}_i(f_i^{t+1}) - \mathcal{L}_i(f_i^t)\right] \\
\leq\ &\sum_{i=1}^d -\eta_i^t \left(3/4 - L_\mathcal{L}^2(\eta_i^t)^2(M_K)^2\right) \cdot \left(m_{i,t}(\mu_i) + \frac{m_{i,t}''(\mu_i)}{2}\sigma_i^2\right).
\end{aligned}
\tag{27}
$$

Hence, if $\forall i \in \mathbb{N}_d, \eta_i^t \leq \frac{1}{2L_{\mathcal{L}} \cdot M_K}$, we have

$$
\begin{aligned}
\mathbb{E}_{\boldsymbol{x} \sim [\mathbb{P}_i(x_i)]^d} \left[ \mathcal{L}(\boldsymbol{f}^{t+1}) - \mathcal{L}(\boldsymbol{f}^t) \right] &\leq -\frac{\tilde{\eta}^t}{2} \sum_{i=1}^d (m_{i,t}(\mu_i) + \frac{m_{i,t}''(\mu_i)}{2} \sigma_i^2) \\
&\leq -\frac{\tilde{\eta}^t d}{2} \cdot \min_{i \in \mathbb{N}_d} \left( m_{i,t}(\mu_i) + \frac{m_{i,t}''(\mu_i)}{2} \sigma_i^2 \right),
\end{aligned}
\tag{28}
$$

where $\tilde{\eta}^t = \min_{i \in \mathbb{N}_d} \eta_i^t$.

∎

**Proof of Theorem 8**    Recall Lemma 7, $\forall i \in \mathbb{N}_d, \eta_i^t \leq \frac{1}{2L_{\mathcal{L}} \cdot M_K}$, we have

$$
\begin{aligned}
\mathbb{E}_{\boldsymbol{x} \sim [\mathbb{P}_i(x_i)]^d} \left[ \mathcal{L}(\boldsymbol{f}^{t+1}) - \mathcal{L}(\boldsymbol{f}^t) \right] &\leq -\frac{\tilde{\eta}^t}{2} \sum_{i=1}^d (m_{i,t}(\mu_i) + \frac{m_{i,t}''(\mu_i)}{2} \sigma_i^2) \\
&\leq -\frac{\tilde{\eta}^t}{2} d \cdot \min_{i \in \mathbb{N}_d} \left( m_{i,t}(\mu_i) + \frac{m_{i,t}''(\mu_i)}{2} \sigma_i^2 \right),
\end{aligned}
\tag{29}
$$

where $\tilde{\eta}^t = \min_{i \in \mathbb{N}_d} \eta_i^t$. Rearranging above, we have:

$$
\frac{2\mathbb{E}_{\boldsymbol{x} \sim [\mathbb{P}_i(x_i)]^d} \left[ \mathcal{L}(\boldsymbol{f}^t) - \mathcal{L}(\boldsymbol{f}^{t+1}) \right]}{d\tilde{\eta}^t} \geq \min_{i \in \mathbb{N}_d} \left( m_{i,t}(\mu_i) + \frac{m_{i,t}''(\mu_i)}{2} \sigma_i^2 \right).
\tag{30}
$$

Equivalently, replace index $t$ by $l$, $\frac{2\mathbb{E}_{\boldsymbol{x} \sim [\mathbb{P}_i(x_i)]^d} \left[ \mathcal{L}(\boldsymbol{f}^l) - \mathcal{L}(\boldsymbol{f}^{l+1}) \right]}{d\tilde{\eta}^l} \geq \min_{i \in \mathbb{N}_d} \left( m_{i,l}(\mu_i) + \frac{m_{i,l}''(\mu_i)}{2} \sigma_i^2 \right)$.
Consequently, plugging $l = 0, 1 \ldots, t-1$ in it and summing them up, we hence have

$$
\begin{aligned}
\sum_{l=0}^{t-1} &\min_{i \in \mathbb{N}_d} \left( m_{i,l}(\mu_i) + \frac{m_{i,l}''(\mu_i)}{2} \sigma_i^2 \right) \\
&\leq \frac{2}{d} \sum_{l=0}^{t-1} \frac{\mathbb{E}_{\boldsymbol{x} \sim [\mathbb{P}_i(x_i)]^d} \left[ \mathcal{L}(\boldsymbol{f}^l) - \mathcal{L}(\boldsymbol{f}^{l+1}) \right]}{\tilde{\eta}^l} \\
&\leq \frac{2}{d\dot{\eta}} \sum_{l=0}^{t-1} \mathbb{E}_{\boldsymbol{x} \sim [\mathbb{P}_i(x_i)]^d} \left[ \mathcal{L}(\boldsymbol{f}^l) - \mathcal{L}(\boldsymbol{f}^{l+1}) \right],
\end{aligned}
\tag{31}
$$

where $\dot{\eta} = \min_{l \in \{0\} \bigcup \mathbb{N}_{t-1}} \tilde{\eta}^l > 0$. Expanding the r.h.s. term in Eq. 31 yields

$$
\begin{aligned}
\frac{2}{d\dot{\eta}} &\sum_{l=0}^{t-1} \mathbb{E}_{\boldsymbol{x} \sim [\mathbb{P}_i(x_i)]^d} \left[ \mathcal{L}(\boldsymbol{f}^l) - \mathcal{L}(\boldsymbol{f}^{l+1}) \right] \\
&= \frac{2}{d\dot{\eta}} \mathbb{E}_{\boldsymbol{x} \sim [\mathbb{P}_i(x_i)]^d} \left[ \mathcal{L}(\boldsymbol{f}^0) - \mathcal{L}(\boldsymbol{f}^t) \right] \\
&\leq \frac{2}{d\dot{\eta}} \mathbb{E}_{\boldsymbol{x} \sim [\mathbb{P}_i(x_i)]^d} \left[ \mathcal{L}(\boldsymbol{f}^0) \right].
\end{aligned}
\tag{32}
$$

In terms of the l.h.s. term in Eq. 31, we must have

$$
\sum_{l=0}^{t-1} \min_{i \in \mathbb{N}_d} \left( m_{i,l}(\mu_i) + \frac{m_{i,l}''(\mu_i)}{2} \sigma_i^2 \right) \geq t \cdot \min_{l \in \{0\} \bigcup \mathbb{N}_{t-1}} \min_{i \in \mathbb{N}_d} \left( m_{i,l}(\mu_i) + \frac{m_{i,l}''(\mu_i)}{2} \sigma_i^2 \right).
\tag{33}
$$

Combining expression 32 and 33, we thus have

$$
\begin{aligned}
t \cdot \min_{l \in \{0\} \bigcup \mathbb{N}_{t-1}} \min_{i \in \mathbb{N}_d} \left( m_{i,l}(\mu_i) + \frac{m_{i,l}''(\mu_i)}{2} \sigma_i^2 \right) &\leq \sum_{l=0}^{t-1} \min_{i \in \mathbb{N}_d} \left( m_{i,l}(\mu_i) + \frac{m_{i,l}''(\mu_i)}{2} \sigma_i^2 \right) \\
&\leq \frac{2}{d\dot{\eta}} \mathbb{E}_{\boldsymbol{x} \sim [\mathbb{P}_i(x_i)]^d} \left[ \mathcal{L}(\boldsymbol{f}^0) \right].
\end{aligned}
\tag{34}
$$

Since $m_{i,t}(\mu_i)$ is monotonically decreasing w.r.t. iteration $t$, we have

$$\min_{l \in \{0\} \bigcup \mathbb{N}_{t-1}} \min_{i \in \mathbb{N}_d} \left( m_{i,l}(\mu_i) + m_{i,l}''(\mu_i)\sigma_i^2/2 \right) = \min_{i \in \mathbb{N}_d} \left( m_{i,t-1}(\mu_i) + m_{i,t-1}''(\mu_i)\sigma_i^2/2 \right).$$

Therefore, we can derive

$$\min_{i \in \mathbb{N}_d} \left( m_{i,t-1}(\mu_i) + \frac{m_{i,t-1}''(\mu_i)}{2}\sigma_i^2 \right) \leq \frac{2}{d\dot{\eta}t} \mathbb{E}_{\boldsymbol{x} \sim [\mathbb{P}_i(x_i)]^d} \left[ \mathcal{L}(\boldsymbol{f}^0) \right], \tag{35}$$

when the returned vector-valued model for multiple learners is $\boldsymbol{f}^t$ and index $t-1$ denotes the last iteration.

On the other hand, it follows from Eq. 32 and the fact that $m_{i,t}(\mu_i)$ is monotonically decreasing w.r.t. iteration $t$ that we have

$$
\begin{aligned}
\mathbb{E}_{\boldsymbol{x} \sim [\mathbb{P}_i(x_i)]^d} \left[ \mathcal{L}(\boldsymbol{f}^t) - \mathcal{L}(\boldsymbol{f}^0) \right] &\leq -\frac{d\dot{\eta}}{2} \sum_{l=0}^{t-1} \min_{i \in \mathbb{N}_d} \left( m_{i,l}(\mu_i) + \frac{m_{i,l}''(\mu_i)}{2}\sigma_i^2 \right) \\
&\leq -\frac{d\dot{\eta}}{2}t \cdot \min_{i \in \mathbb{N}_d} \left( m_{i,t-1}(\mu_i) + \frac{m_{i,t-1}''(\mu_i)}{2}\sigma_i^2 \right). 
\end{aligned} \tag{36}
$$

After rearranging, we obtain

$$
\begin{aligned}
\min_{i \in \mathbb{N}_d} \left( m_{i,t-1}(\mu_i) + \frac{m_{i,t-1}''(\mu_i)}{2}\sigma_i^2 \right) &\leq \frac{2}{d\dot{\eta}t} \cdot \mathbb{E}_{\boldsymbol{x} \sim [\mathbb{P}_i(x_i)]^d} \left[ \mathcal{L}(\boldsymbol{f}^0) - \mathcal{L}(\boldsymbol{f}^t) \right] \\
&\leq \frac{2}{d\dot{\eta}t} \cdot \mathbb{E}_{\boldsymbol{x} \sim [\mathbb{P}_i(x_i)]^d} \left[ \mathcal{L}(\boldsymbol{f}^0) \right]
\end{aligned} \tag{37}
$$

Let r.h.s. of Eq. 37 be controlled by a small constant $\epsilon > 0$, we have

$$t \geq \frac{2}{d\dot{\eta}\epsilon} \cdot \mathbb{E}_{\boldsymbol{x} \sim [\mathbb{P}_i(x_i)]^d} \left[ \mathcal{L}(\boldsymbol{f}^0) \right], \tag{38}$$

which means given a small constant $\epsilon > 0$ it would take approximately

$$\frac{2}{d\dot{\eta}\epsilon} \cdot \mathbb{E}_{\boldsymbol{x} \sim [\mathbb{P}_i(x_i)]^d} \left[ \mathcal{L}(\boldsymbol{f}^0) \right]$$

iterations to reach a stationary point.

Additionally, this suggests that multiple learners could achieve the stationary state as: In each iteration, check if $m_{i,t-1}(\mu_i) + \frac{m_{i,t-1}''(\mu_i)}{2}\sigma_i^2$ is small enough. Assuming this condition is satisfied, the learners will have already reduced the multi-learner loss to an acceptably low level, allowing them to send a signal indicating termination back to the teachers. If the condition is not fulfilled, the teachers will continue with the process. The termination occurs within $\frac{2}{d\dot{\eta}\epsilon} \cdot \mathbb{E}_{\boldsymbol{x} \sim [\mathbb{P}_i(x_i)]^d} \left[ \mathcal{L}(\boldsymbol{f}^0) \right]$ iterations.

∎

**Proof of Lemma 9**    Recall practical Greedy Functional Teaching in Eq. 12

$$\left( \boldsymbol{x}^{t*} = \arg\max_{\boldsymbol{x}^t \in \mathcal{X}} \left| E_{\boldsymbol{x}^t} \left[ \nabla_f \mathcal{L}(f)|_{f=f^t} \right] \right|, y^* = E_{\boldsymbol{x}^{t*}} [f^*] \right). \tag{39}$$

Obviously, it is trivial to see that $\forall \boldsymbol{x}^t \in \mathcal{X}$,

$$\left| E_{\boldsymbol{x}^{t*}} \left[ \nabla_f \mathcal{L}(f)|_{f=f^t} \right] \right|^2 \geq \left| E_{\boldsymbol{x}^t} \left[ \nabla_f \mathcal{L}(f)|_{f=f^t} \right] \right|^2. \tag{40}$$

Analogous to the Proof of Lemma 7 in B, we can derive

$$
\begin{aligned}
&\mathbb{E}_{x_i \sim \mathbb{P}_i(x_i)} \left[ \mathcal{L}_i(f_i^{t+1}) - \mathcal{L}_i(f_i^t) \right] \\
\leq{}& -\eta_i^t \cdot \mathbb{E}_{x_i \sim \mathbb{P}_i(x_i)} \left[ E_{x_i^{t*}} \left[ \nabla_f \mathcal{L}_i(f)|_{f=f_i^t} \cdot \nabla_f \mathcal{L}_i(f)|_{f=f_i^{t+1}} \right] \right] \\
\overset{*}{=}{}& -\eta_i^t \cdot E_{x_i^{t*}} \left[ \nabla_f \mathcal{L}_i(f)|_{f=f_i^t} \cdot \nabla_f \mathcal{L}_i(f)|_{f=f_i^{t+1}} \right] \\
\leq{}& -\eta_i^t \cdot \left( 3/4 - L_{\mathcal{L}}^2(\eta_i^t)^2 (M_K)^2 \right) E_{x_i^{t*}} \left[ \nabla_f^2 \mathcal{L}_i(f)|_{f=f_i^t} \right] \\
={}& -\eta_i^t \cdot \left( 3/4 - L_{\mathcal{L}}^2(\eta_i^t)^2 (M_K)^2 \right) m_{i,t}(x_i^{t*}),
\end{aligned} \tag{41}
$$

where $\overset{*}{=}$ holds because selected $x_i^{t*}$ by GFT is determined. Therefore,

$$\mathbb{E}_{\boldsymbol{x} \sim [\mathbb{P}_i(x_i)]^d} \left[ \mathcal{L}(\boldsymbol{f}^{t+1}) - \mathcal{L}(\boldsymbol{f}^t) \right] \leq \sum_{i=1}^d -\eta_i^t \left( 3/4 - L_{\mathcal{L}}^2 (\eta_i^t)^2 (M_K)^2 \right) m_{i,t}(x_i^{t*}). \tag{42}$$

Hence, if $\forall i \in \mathbb{N}_d, \eta_i^t \leq \frac{1}{2L_{\mathcal{L}} \cdot M_K}$, we have

$$\mathbb{E}_{\boldsymbol{x} \sim [\mathbb{P}_i(x_i)]^d} \left[ \mathcal{L}(\boldsymbol{f}^{t+1}) - \mathcal{L}(\boldsymbol{f}^t) \right] \leq -\frac{\tilde{\eta}^t}{2} \sum_{i=1}^d m_{i,t}(x_i^{t*}) \leq -\frac{\tilde{\eta}^t d}{2} \cdot \min_{i \in \mathbb{N}_d} m_{i,t}(x_i^{t*}), \tag{43}$$

where $\tilde{\eta}^t = \min_{i \in \mathbb{N}_d} \eta_i^t$. Note that GFT is to select examples such that the gradient is the steepest, thus $\min_{i \in \mathbb{N}_d} m_{i,t}(x_i^{t*}) \geq \min_{i \in \mathbb{N}_d} \left( m_{i,t}(\mu_i) + \frac{m_{i,t}''(\mu_i)}{2} \sigma_i^2 \right)$

∎

**Proof of Theorem 10**   Recall the result of Lemma 7, when $\forall i \in \mathbb{N}_d, \eta_i^t \leq \frac{1}{2L_{\mathcal{L}} \cdot M_K}$

$$\sum_{i=1}^d (m_{i,t}(\mu_i) + \frac{m_{i,t}''(\mu_i)}{2} \sigma_i^2) \leq \frac{2\mathbb{E}_{\boldsymbol{x} \sim [\mathbb{P}_i(x_i)]^d} \left[ \mathcal{L}(\boldsymbol{f}^t) - \mathcal{L}(\boldsymbol{f}^{t+1}) \right]}{\tilde{\eta}^t}, \tag{44}$$

where $\tilde{\eta}^t = \min_{i \in \mathbb{N}_d} \eta_i^t$.

Before converging to the stationary state, $\sum_{i=1}^d m_{i,t}(x_i^{t*}) > 0$. Therefore, we can express it as

$$\sum_{i=1}^d m_{i,t}(x_i^{t*}) \cdot \frac{\sum_{i=1}^d (m_{i,t}(\mu_i) + \frac{m_{i,t}''(\mu_i)}{2} \sigma_i^2)}{\sum_{i=1}^d m_{i,t}(x_i^{t*})} \leq \frac{2\mathbb{E}_{\boldsymbol{x} \sim [\mathbb{P}_i(x_i)]^d} \left[ \mathcal{L}(\boldsymbol{f}^t) - \mathcal{L}(\boldsymbol{f}^{t+1}) \right]}{\tilde{\eta}^t}. \tag{45}$$

We see that $\frac{\sum_{i=1}^d (m_{i,t}(\mu_i) + \frac{m_{i,t}''(\mu_i)}{2} \sigma_i^2)}{\sum_{i=1}^d m_{i,t}(x_i^{t*})} \leq 1$ measures the difference between two algorithms at $t$ iteration. And this is deterministic in each iteration, that is, this can be estimated before sampling, so we can see the superiority of GFT.

$$\begin{aligned}
&\frac{\sum_{i=1}^d (m_{i,t}(\mu_i) + \frac{m_{i,t}''(\mu_i)}{2} \sigma_i^2)}{\sum_{i=1}^d m_{i,t}(x_i^{t*})} \\
&= 1 - \frac{\sum_{i=1}^d \left( m_{i,t}(x_i^{t*}) - m_{i,t}(\mu_i) - \frac{m_{i,t}''(\mu_i)}{2} \sigma_i^2 \right)}{\sum_{i=1}^d m_{i,t}(x_i^{t*})} \\
&\geq 1 - \frac{\sum_{i=1}^d \left( \|x_i^{t*} - \mu_i\|_2 - \frac{m_{i,t}''(\mu_i)}{2} \sigma_i^2 \right)}{\sum_{i=1}^d m_{i,t}(x_i^{t*})} \\
&\geq 1 - \frac{\sum_{i=1}^d \left( \|x_i^{t*} - \mu_i\|_2 \right)}{\sum_{i=1}^d m_{i,t}(x_i^{t*})}
\end{aligned} \tag{46}$$

where we assume $m_{i,t}(\dot{x})$ is Lipschitz continuous w.r.t. input $\dot{x}$ to tract the relation between this quantity and the distance between $\mu_i$ and $x_i^{t*}$. Now, we have

$$1 - \frac{\sum_{i=1}^d \left( \|x_i^{t*} - \mu_i\|_2 \right)}{\sum_{i=1}^d m_{i,t}(x_i^{t*})} \leq \frac{\sum_{i=1}^d (m_{i,t}(\mu_i) + \frac{m_{i,t}''(\mu_i)}{2} \sigma_i^2)}{\sum_{i=1}^d m_{i,t}(x_i^{t*})} \leq 1. \tag{47}$$

We see that when selected $x_i^{t*}$ by GFT is close to $\mu_i$ then RFT and GFT on average share the same performance. (show that if maximal model disagreement occurs at the mean of $x$ distribution, then such a greedy teacher may share similar performance with a random teacher on average.) This is important to gain an insight on when a greedy teacher is better than a random teacher. Then, we have

$$\begin{aligned}
\sum_{i=1}^d m_{i,t}(x_i^{t*}) - \sum_{i=1}^d \left( \|x_i^{t*} - \mu_i\|_2 \right) &= \sum_{i=1}^d \left( m_{i,t}(x_i^{t*}) - \|x_i^{t*} - \mu_i\|_2 \right) \\
&\leq \frac{2\mathbb{E}_{\boldsymbol{x} \sim [\mathbb{P}_i(x_i)]^d} \left[ \mathcal{L}(\boldsymbol{f}^t) - \mathcal{L}(\boldsymbol{f}^{t+1}) \right]}{\tilde{\eta}^t}.
\end{aligned} \tag{48}$$

$$\sum_{i=1}^{d} m_{i,t}(x_i^{t^*}) \leq \frac{2\mathbb{E}_{\boldsymbol{x}\sim[\mathbb{P}_i(x_i)]^d}\left[\mathcal{L}(\boldsymbol{f}^t) - \mathcal{L}(\boldsymbol{f}^{t+1})\right]}{\tilde{\eta}^t} + \sum_{i=1}^{d}\left(\|x_i^{t^*} - \mu_i\|_2\right). \tag{49}$$

Therefore,

$$\frac{2\mathbb{E}_{\boldsymbol{x}\sim[\mathbb{P}_i(x_i)]^d}\left[\mathcal{L}(\boldsymbol{f}^t) - \mathcal{L}(\boldsymbol{f}^{t+1})\right]}{d\tilde{\eta}^t} + \frac{1}{d}\sum_{i=1}^{d}\left(\|x_i^{t^*} - \mu_i\|_2\right) \geq \min_{i\in\mathbb{N}_d} m_{i,t}(x_i^{t^*}). \tag{50}$$

Equivalently, replace index $t$ by $l$, $\frac{2\mathbb{E}_{\boldsymbol{x}\sim[\mathbb{P}_i(x_i)]^d}\left[\mathcal{L}(\boldsymbol{f}^l) - \mathcal{L}(\boldsymbol{f}^{l+1})\right]}{d\tilde{\eta}^l} + \frac{1}{d}\sum_{i=1}^{d}\left(\|x_i^{l^*} - \mu_i\|_2\right) \geq$ $\min_{i\in\mathbb{N}_d} m_{i,l}(x_i^{l^*})$. Consequently, plugging $l = 0, 1 \ldots, t-1$ in it and summing them up, we hence have

$$\sum_{l=0}^{t-1} \min_{i\in\mathbb{N}_d} m_{i,l}(x_i^{l^*})$$

$$\leq \frac{2}{d}\sum_{l=0}^{t-1} \frac{\mathbb{E}_{\boldsymbol{x}\sim[\mathbb{P}_i(x_i)]^d}\left[\mathcal{L}(\boldsymbol{f}^l) - \mathcal{L}(\boldsymbol{f}^{l+1})\right]}{\tilde{\eta}^l} + \frac{1}{d}\sum_{l=0}^{t-1}\sum_{i=1}^{d}\left(\|x_i^{l^*} - \mu_i\|_2\right)$$

$$\leq \frac{2}{d\dot{\eta}}\sum_{l=0}^{t-1} \mathbb{E}_{\boldsymbol{x}\sim[\mathbb{P}_i(x_i)]^d}\left[\mathcal{L}(\boldsymbol{f}^l) - \mathcal{L}(\boldsymbol{f}^{l+1})\right] + \frac{1}{d}\sum_{l=0}^{t-1}\sum_{i=1}^{d}\left(\|x_i^{l^*} - \mu_i\|_2\right), \tag{51}$$

where $\dot{\eta} = \min_{l\in\{0\}\bigcup\mathbb{N}_{t-1}} \tilde{\eta}^l > 0$. Expanding the r.h.s. term in Eq. 51 yields

$$\frac{2}{d\dot{\eta}}\sum_{l=0}^{t-1} \mathbb{E}_{\boldsymbol{x}\sim[\mathbb{P}_i(x_i)]^d}\left[\mathcal{L}(\boldsymbol{f}^l) - \mathcal{L}(\boldsymbol{f}^{l+1})\right] + \frac{1}{d}\sum_{l=0}^{t-1}\sum_{i=1}^{d}\left(\|x_i^{l^*} - \mu_i\|_2\right)$$

$$= \frac{2}{d\dot{\eta}}\mathbb{E}_{\boldsymbol{x}\sim[\mathbb{P}_i(x_i)]^d}\left[\mathcal{L}(\boldsymbol{f}^0) - \mathcal{L}(\boldsymbol{f}^t)\right] + \frac{1}{d}\sum_{l=0}^{t-1}\sum_{i=1}^{d}\left(\|x_i^{l^*} - \mu_i\|_2\right)$$

$$\leq \frac{2}{d\dot{\eta}}\mathbb{E}_{\boldsymbol{x}\sim[\mathbb{P}_i(x_i)]^d}\left[\mathcal{L}(\boldsymbol{f}^0)\right] + \frac{1}{d}\sum_{l=0}^{t-1}\sum_{i=1}^{d}\left(\|x_i^{l^*} - \mu_i\|_2\right). \tag{52}$$

In terms of the l.h.s. term in Eq. 51, we must have

$$\sum_{l=0}^{t-1} \min_{i\in\mathbb{N}_d} m_{i,l}(x_i^{l^*}) \geq t \cdot \min_{l\in\{0\}\bigcup\mathbb{N}_{t-1}} \min_{i\in\mathbb{N}_d} m_{i,l}(x_i^{l^*}). \tag{53}$$

Combining expression 51, 52 and 53, we thus have

$$t \cdot \min_{l\in\{0\}\bigcup\mathbb{N}_{t-1}} \min_{i\in\mathbb{N}_d} m_{i,l}(x_i^{l^*}) \leq \sum_{l=0}^{t-1} \min_{i\in\mathbb{N}_d} m_{i,l}(x_i^{l^*})$$

$$\leq \frac{2}{d\dot{\eta}}\mathbb{E}_{\boldsymbol{x}\sim[\mathbb{P}_i(x_i)]^d}\left[\mathcal{L}(\boldsymbol{f}^0)\right] + \frac{1}{d}\sum_{l=0}^{t-1}\sum_{i=1}^{d}\left(\|x_i^{l^*} - \mu_i\|_2\right). \tag{54}$$

Since $m_{i,t}(x_i^{t^*})$ is monotonically non-increasing w.r.t. iteration $t$, we can derive

$$\min_{i\in\mathbb{N}_d} m_{i,t-1}(x_i^{t-1^*}) \leq \frac{2}{d\dot{\eta}t}\mathbb{E}_{\boldsymbol{x}\sim[\mathbb{P}_i(x_i)]^d}\left[\mathcal{L}(\boldsymbol{f}^0)\right] + \frac{1}{dt}\sum_{l=0}^{t-1}\sum_{i=1}^{d}\left(\|x_i^{l^*} - \mu_i\|_2\right), \tag{55}$$

when the returned vector-valued model for multiple learners is $\boldsymbol{f}^t$.

To compare with RFT, we can plug the r.h.s. of Eq. 38 and see that the loss reduction is more than $\mathbb{E}_{\boldsymbol{x}\sim[\mathbb{P}_i(x_i)]^d}\left[\mathcal{L}(\boldsymbol{f}^0)\right]$, which indicates GFT needs less iterations to converge and the efficiency of GFT. Compared with Eq. 35 and 55, we see that the inferiority of RFT compared to GFT comes from the cumulative distance between the example mean and the example selected by GFT.

■

**Proof of Proportion 11**  When the proximity between $\boldsymbol{f}^t$ and $\boldsymbol{f}^*$ is sufficiently close, we have $\epsilon < \epsilon_0$ where $\epsilon_0$ is the pre-defined approximation error of employed two-layer perceptron. Take $A = I_d$, then we have $\|A \cdot \boldsymbol{f}^t - \boldsymbol{f}^*\|_{\mathcal{H}^d} = \|I_d \cdot \boldsymbol{f}^t - \boldsymbol{f}^*\|_{\mathcal{H}^d} = \|\boldsymbol{f}^t - \boldsymbol{f}^*\|_{\mathcal{H}^d} \leq \epsilon \leq \epsilon_0$, which means that this perceptron have searched the matrix $A = I_d$ that satisfies $\|A \cdot \boldsymbol{f}^t - \boldsymbol{f}^*\|_{\mathcal{H}^d} \leq \epsilon_0$.

■

**Proof of Lemma 12**  Since the multi-learner convex loss $\mathcal{L}(\boldsymbol{f})$ will decrease as input $\boldsymbol{f}$ close to $\boldsymbol{f}^*$ and matrix $A^t$ comes from $A^t = \arg\min_{A \in \mathbb{R}^{d \times d}} \|A\boldsymbol{f}^t - \boldsymbol{f}^*\|_{\mathcal{H}^d}$, searching a matrix $A$ to minimize the disagreement between $A\boldsymbol{f}^t$ and $\boldsymbol{f}^*$, it is trivial to see that

$$\mathcal{L}(\boldsymbol{f}^t) - \mathcal{L}(A^t \boldsymbol{f}^t) \geq 0. \tag{56}$$

Based on Assumption 3, we can derive that

$$|\mathcal{L}(A^t \boldsymbol{f}^t) - \mathcal{L}(\boldsymbol{f}^*)| \leq L_{\mathcal{L}} \|A^t \boldsymbol{f}^t - \boldsymbol{f}^*\|_{\mathcal{H}^d} \tag{57}$$

and

$$|\mathcal{L}(\boldsymbol{f}^t) - \mathcal{L}(\boldsymbol{f}^*)| \leq L_{\mathcal{L}} \|\boldsymbol{f}^t - \boldsymbol{f}^*\|_{\mathcal{H}^d}. \tag{58}$$

Therefore, we have

$$
\begin{aligned}
\mathcal{L}(\boldsymbol{f}^t) - \mathcal{L}(A^t \boldsymbol{f}^t) &= \mathcal{L}(\boldsymbol{f}^t) - \mathcal{L}(\boldsymbol{f}^*) + \mathcal{L}(\boldsymbol{f}^*) - \mathcal{L}(A^t \boldsymbol{f}^t) \\
&\leq |\mathcal{L}(\boldsymbol{f}^t) - \mathcal{L}(\boldsymbol{f}^*)| + |\mathcal{L}(\boldsymbol{f}^*) - \mathcal{L}(A^t \boldsymbol{f}^t)| \\
&\leq L_{\mathcal{L}} \cdot (\|A^t \boldsymbol{f}^t - \boldsymbol{f}^*\|_{\mathcal{H}^d} + \|\boldsymbol{f}^t - \boldsymbol{f}^*\|_{\mathcal{H}^d}) \\
&\leq 2L_{\mathcal{L}} \cdot \|\boldsymbol{f}^t - \boldsymbol{f}^*\|_{\mathcal{H}^d},
\end{aligned} \tag{59}
$$

which concludes the proof.

■

**Proof of Theorem 13**  Following the style of the previous proof, *e.g.*, the proof of Lemma 7, we begin by investigating the reduction in the loss for a single learner.

$$
\begin{aligned}
& \mathbb{E}_{x_i \sim \mathbb{P}_i(x_i)} \left[ \mathcal{L}_i(f_{A^t,i}^{t+1}) - \mathcal{L}_i(f_i^t) \right] \\
=\ & \mathbb{E}_{x_i \sim \mathbb{P}_i(x_i)} \left[ \mathcal{L}_i(f_{A^t,i}^{t+1}) - \mathcal{L}_i(A_{(i,\cdot)}^t \boldsymbol{f}^t) + \mathcal{L}_i(A_{(i,\cdot)}^t \boldsymbol{f}^t) - \mathcal{L}_i(f_i^t) \right] \\
=\ & \mathbb{E}_{x_i \sim \mathbb{P}_i(x_i)} \left[ \mathcal{L}_i(f_{A^t,i}^{t+1}) - \mathcal{L}_i(A_{(i,\cdot)}^t \boldsymbol{f}^t) \right] + \mathbb{E}_{x_i \sim \mathbb{P}_i(x_i)} \left[ \mathcal{L}_i(A_{(i,\cdot)}^t \boldsymbol{f}^t) - \mathcal{L}_i(f_i^t) \right] \\
\overset{*}{\leq}\ & \mathbb{E}_{x_i \sim \mathbb{P}_i(x_i)} \left[ \mathcal{L}_i(f_{A^t,i}^{t+1}) - \mathcal{L}_i(A_{(i,\cdot)}^t \boldsymbol{f}^t) \right],
\end{aligned} \tag{60}
$$

where it follows from Lemma 12 that $\overset{*}{\leq}$ holds. Therefore, we have

$$
\begin{aligned}
& \mathbb{E}_{\boldsymbol{x} \sim [\mathbb{P}_i(x_i)]^d} \left[ \mathcal{L}(\boldsymbol{f}_{A^t}^{t+1}) - \mathcal{L}(\boldsymbol{f}^t) \right] \\
=\ & \mathbb{E}_{\boldsymbol{x} \sim [\mathbb{P}_i(x_i)]^d} \left[ \sum_{i=1}^{d} \mathcal{L}_i(f_{A^t,i}^{t+1}) - \sum_{i=1}^{d} \mathcal{L}_i(f_i^t) \right] \\
=\ & \sum_{i=1}^{d} \mathbb{E}_{x_i \sim \mathbb{P}_i(x_i)} \left[ \mathcal{L}_i(f_{A^t,i}^{t+1}) - \mathcal{L}_i(f_i^t) \right] \\
\leq\ & \sum_{i=1}^{d} \mathbb{E}_{x_i \sim \mathbb{P}_i(x_i)} \left[ \mathcal{L}_i(f_{A^t,i}^{t+1}) - \mathcal{L}_i(A_{(i,\cdot)}^t \boldsymbol{f}^t) \right] \\
=\ & \mathbb{E}_{\boldsymbol{x} \sim [\mathbb{P}_i(x_i)]^d} \left[ \mathcal{L}(\boldsymbol{f}_{A^t}^{t+1}) - \mathcal{L}(A^t \boldsymbol{f}^t) \right] \leq 0,
\end{aligned} \tag{61}
$$

which completes the proof.

■

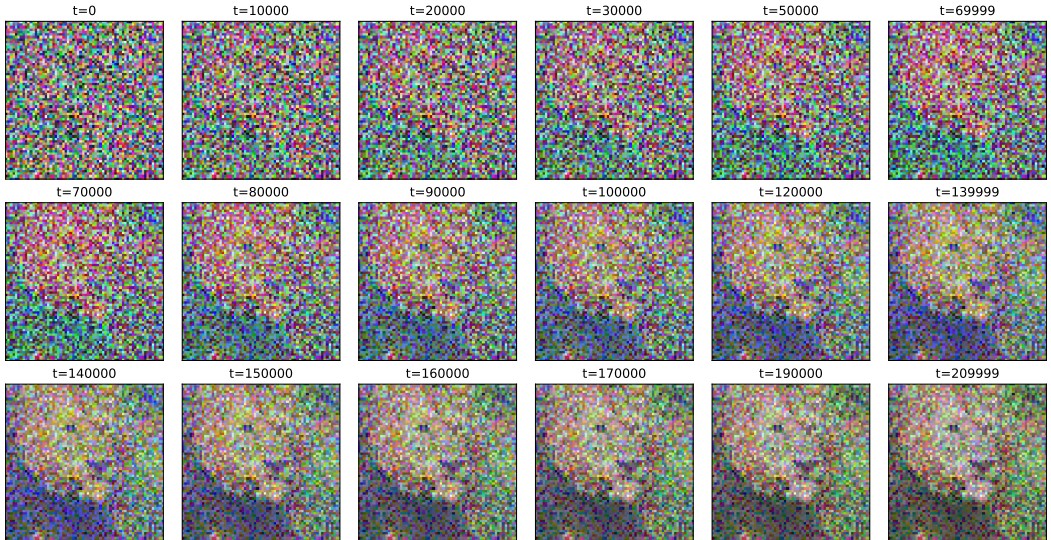

Figure 6: Extensive visualization of $\boldsymbol{f}^t$ for single-learner teaching.

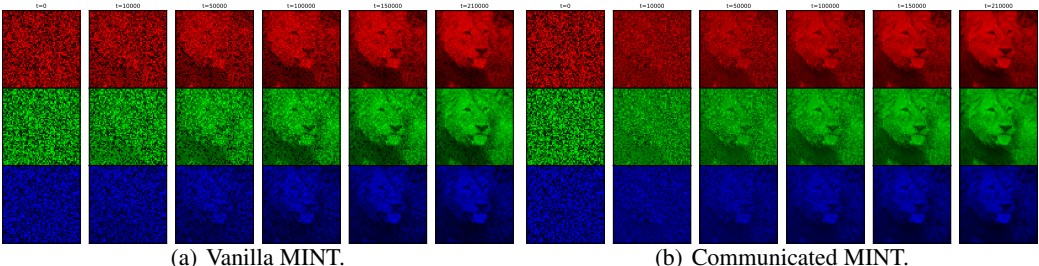

(a) Vanilla MINT.                    (b) Communicated MINT.

Figure 7: The channel-wise visualization of specific $\boldsymbol{f}^t$ corresponding to Figure 4 (b)-(c).

## C   Detailed Experiments and Extensions

Since computers operate in a discrete manner, we employ dense pairwise points $\{(x_i, f(x_i))\}_{i \in \mathbb{N}_n}$ to represent a scalar-valued function $f$ and points $\{([x_i]_j^d, [f_i(x_i)]_j^d)\}_{j \in \mathbb{N}_m}$ to represent a vector-valued function $\boldsymbol{f} = [f_i]^d \in \mathcal{H}^d$. To facilitate visualization, our experiments have utilized 2D (1D Gaussian data), 3D (a grayscale image) and 4D (a colored image) examples to demonstrate the insights obtained from our theoretical analysis. Generally, the domain of the functions being learned in 3D cases is determined by the $x$ and $y$ values, which represent the pixel locations, while the range is represented by the $z$ values, indicating the color levels. This is comparable for 2D and 4D cases. Besides, for high-dimensional vision datasets that can be formulated as vector-valued functions, the methodology developed in this work can be applied as well. For all experiments, we align with [73] to set RBF $K(x, x') = \exp\left(-\left\|\frac{x-x'}{2}\right\|_2^2\right)$ as the kernel and to take empirical (average) $L_2$ norm defined in vector-valued Hilbert space to measure the difference between $\boldsymbol{f} \in \mathcal{H}^d$ and $\boldsymbol{f}^* \in \mathcal{H}^d$,

$$\mathcal{M}(\boldsymbol{f}, \boldsymbol{f}^*) = \|\boldsymbol{f} - \boldsymbol{f}^*\|_{\mathcal{H}^d} = \frac{1}{dn}\sqrt{\sum_{i=1}^{d}\sum_{j=1}^{n}\left(f_i(x_{i,j}) - f_i^*(x_{i,j})\right)^2}.$$

Our implementation relies on the Intel(R) Core(TM) i7-8750H processor and utilizes NVIDIA graphics cards, specifically the GTX 1050 Ti with Max-Q Design and RTX6000.

**MINT in gray scale.** For impartation of a tiger[5] and a cheetah [15], we assume the loss functions for both learners are square loss $\mathcal{L}_i = (y - f_i(x))^2$, $i \in \mathbb{N}_2$. For the monochrome lion divided into $4 \times 4$

---

[5]

Table 1: Peak signal-to-noise ratio (PSNR) corresponding to learnt $\boldsymbol{f}^t$ show in Figure 4.

|  | 0 | 10,000 | 50,000 | 100,000 | 150,000 | 210,000 |
|---|---|---|---|---|---|---|
| Single-learner teaching | 8.73 | 9.16 | 10.23 | 11.66 | 12.99 | 14.49 |
| Vanilla MINT | 8.73 | 10.01 | 13.19 | 16.27 | 18.98 | 22.00 |
| Communicated MINT | 8.73 | 12.38 | 15.62 | 18.74 | 21.50 | 24.58 |

Table 2: PSNR corresponding to learnt $\boldsymbol{f}^t$ show in Figure 8.

|  | 0 | 10,000 | 50,000 | 100,000 | 150,000 | 210,000 |
|---|---|---|---|---|---|---|
| Single-learner teaching | 8.73 | 8.87 | 9.45 | 10.17 | 10.89 | 11.75 |
| Vanilla MINT | 8.73 | 9.16 | 10.88 | 13.03 | 15.19 | 17.76 |
| Communicated MINT | 8.73 | 11.46 | 13.20 | 15.39 | 17.58 | 20.19 |

pieces, each piece is learnt by a single learner whose loss function is set to be square loss, and all loss functions are the same, $\mathcal{L}_i = (y - f_i(x))^2$, $i \in \mathbb{N}_{16}$. This enables the teaching of a single-learner target model having large input spaces.

**MINT in three (RGB) channels.** Three learners, each with their own randomly-initialized $f^0$, are tasked with learning the three (RGB) channel lion image[6]. Each learner is equipped with a square loss function to learn their respective channels. In the case of single-learner teaching, the teacher must repeatedly teach all three channels. In Vanilla MINT, the teacher can teach all three channels concurrently. In communicated MINT, the teacher uses a two-layer perceptron to solve for the communication matrix $A^t$, with the linear layer initialized by an identity matrix. This allows the teacher to teach all three channels simultaneously.

We present a comprehensive visualization of $\boldsymbol{f}^t$ for Figure 4 (a) in Figure 6, while the channel-specific visualization of certain $\boldsymbol{f}^t$ corresponding to Figure 4 (b)-(c) is exhibited in Figure 7. In order to evaluate the quality of the learned $\boldsymbol{f}^t$ (as images), we utilize the Peak Signal-to-Noise Ratio (PSNR), and the results corresponding to Figure 4 are presented in Table 1. Meanwhile, Figure 8 displays the specific learned $\boldsymbol{f}^t$ for RFT, and its corresponding PSNR is listed in Table 2. Additionally, Figures 9 (a) and (b) respectively exhibit the channel-specific visualizations of vanilla (Figure 8 (b)) and communicated (Figure 8 (c)) MINT for RFT.

In addition, we also investigate the performance of single-learner teaching, as well as vanilla and communicated MINT under a specific initialization of $\boldsymbol{f}^0$, which is obtained through a linear combination of $\boldsymbol{f}^*$ using the inverse of matrix

$$\begin{pmatrix} 1.26 & 2.22 & 3.60 \\ 2.47 & -0.53 & 2.36 \\ 2.40 & 1.68 & 0.40 \end{pmatrix}.$$

Figure 10 displays the plot of the loss. Figure 11 showcases the specific learned $\boldsymbol{f}^t$ for three versions of GFT throughout each iteration. From Figure 11, we can observe that MINT consistently outperforms the single-learner teaching, and additionally, communicated MINT exhibits better performance compared to vanilla MINT. Figure 12 presents the channel-specific visualization of certain $\boldsymbol{f}^t$ that correspond to Figure 11 (b)-(c), while Table 3 lists the PSNR for all comparable $\boldsymbol{f}^t$ in Figure 11. Additionally, Figure 13 displays the specific learned $\boldsymbol{f}^t$ for RFT, while the channel-specific visualization of certain $\boldsymbol{f}^t$ for MINT is exhibited in Figure 14. The corresponding PSNR values are

---

[6]https://bmild.github.io/fourfeat/img/lion_orig.png

Table 3: PSNR corresponding to learnt $\boldsymbol{f}^t$ show in Figure 11.

|  | 0 | 2,000 | 10,000 | 20,000 | 50,000 | 90,000 |
|---|---|---|---|---|---|---|
| Single-learner teaching | 7.58 | 7.65 | 7.88 | 8.15 | 8.88 | 10.07 |
| Vanilla MINT | 7.58 | 7.82 | 8.56 | 9.35 | 11.37 | 13.68 |
| Communicated MINT | 7.58 | 16.29 | 17.12 | 17.98 | 20.19 | 22.75 |

Table 4: PSNR corresponding to learnt $\boldsymbol{f}^t$ show in Figure 13.

|  | 0 | 2,000 | 10,000 | 20,000 | 50,000 | 90,000 |
|---|---|---|---|---|---|---|
| Single-learner teaching | 7.58 | 7.60 | 7.72 | 7.87 | 8.33 | 8.92 |
| Vanilla MINT | 7.58 | 7.66 | 8.03 | 8.48 | 9.82 | 11.62 |
| Communicated MINT | 7.58 | 16.08 | 16.44 | 16.89 | 18.23 | 20.03 |

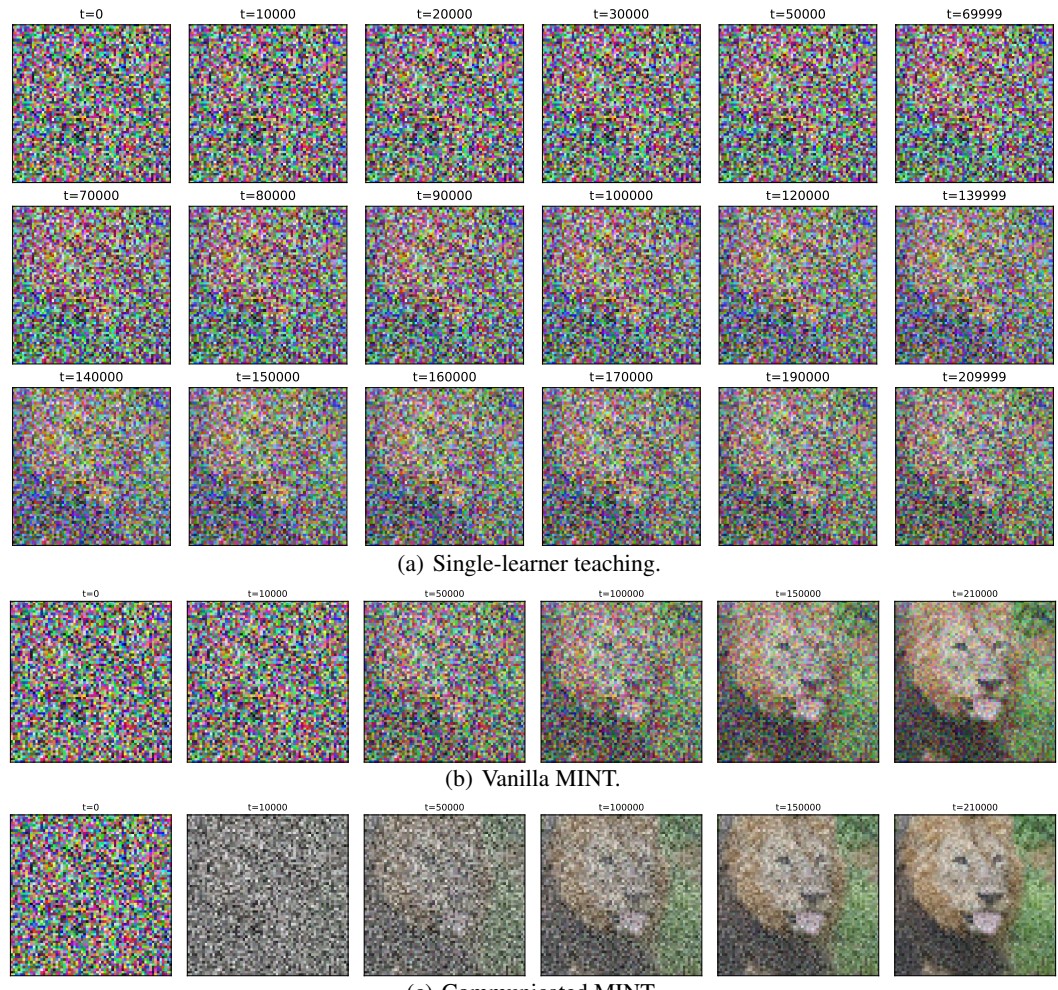

(a) Single-learner teaching.

(b) Vanilla MINT.

(c) Communicated MINT.

Figure 8: Visualization of $\boldsymbol{f}^t$ taught by RFT under the random initialization.

listed in Table 4. An interesting finding is that with this specific initialization, the teacher can assist the learners in directly learning $\boldsymbol{f}^*$ by providing the communication matrix

$$\begin{pmatrix} 1.27 & 2.22 & 3.60 \\ 2.47 & -0.53 & 2.36 \\ 2.40 & 1.68 & 0.40 \end{pmatrix},$$

which is solved by a two-layer perceptron. In order to elaborate this observation, we offer an illustration in Figure 15.

**Synthetic 1D Gaussian data.** We use a synthetic 1D Gaussian data set for teaching a single learner to demonstrate that the greater the distance between the example mean and the example selected by GFT, the more significant the difference between RFT and GFT. We set $f^0 = 0$ and $f^* = \mathcal{N}(x; 0, 5^2)$, where $\mathcal{N}(x; \mu, \sigma^2)$ represents the probability density function of a Gaussian distribution with mean $\mu$ and standard deviation $\sigma$. It is trivial to see that GFT will select examples primarily around $x = 0$. Looking at the loss depicted in Figure 16 (a), we observe that GFT reduces the loss more rapidly than

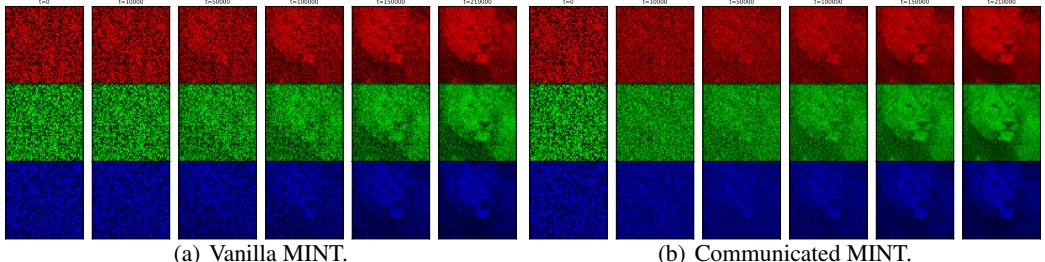

|(a) Vanilla MINT.|(b) Communicated MINT.|

Figure 9: The channel-wise visualization of specific $\boldsymbol{f}^t$ corresponding to Figure 8 (b)-(c).

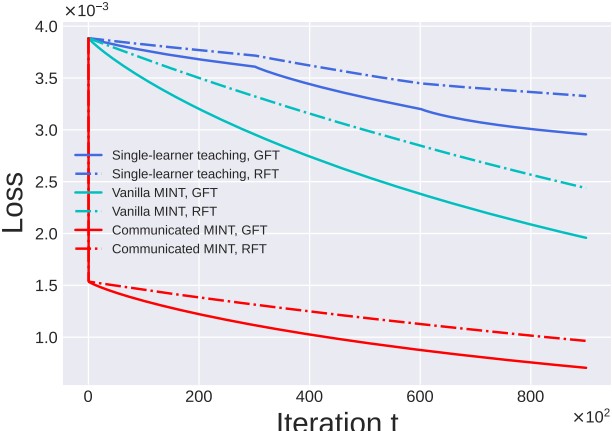

Figure 10: Convergence performance for teaching RGB lion under a particular initialization of $\boldsymbol{f}^0$.

RFT in the settings of $\mu = 0, \mu = -4, \mu = -7$ (in an order of decreasing speed). This is consistent with the observation in Theorem 10. Additionally, we present the specific learned function $f^t$ in Figure 16 (b).

**Synthetic bivariate mixture Gaussian data.** To further showcase the effectiveness of MINT, we utilize synthetic bivariate mixture Gaussian data. It is well-established that a bivariate Gaussian distribution $f_{X_1,X_2}(x_1, x_2)$ can be factored as the product of two independent univariate Gaussian distributions $f_{X_1,X_2}(x_1, x_2) = f_{X_1}(x_1)f_{X_2}(x_2)^7$ [8], regardless of whether $X_1$ and $X_2$ are correlated. For simplicity, our primary focus is on the scenario where there is no correlation ($\rho = 0$), and on GFT. We set

$$f_{X_1,X_2}^0(x_1, x_2) = f_{X_1}^0(x_1)f_{X_2}^0(x_2) = \mathcal{N}(x; -2, 1.5^2)\mathcal{N}(x; 2, 1^2)$$

and

$$f_{X_1,X_2}^*(x_1, x_2) = f_{X_1}^*(x_1)f_{X_2}^*(x_2)$$
$$= \left(\mathcal{N}(x; -2, 1.5^2)/3 + 2\mathcal{N}(x; 2, 1^2)/3\right)\left(3\mathcal{N}(x; -2, 1.5^2)/4 + \mathcal{N}(x; 2, 1^2)/4\right).$$

Figure 17 depicts the visualization of $f_{X_1,X_2}^*(x_1, x_2)$. One can also formulate such a single-learner target function into a multi-learner (vector-valued) one $\boldsymbol{f}$, with the first component being $f_{X_1}(x_1)$ and the second component being $f_{X_2}(x_2)$. Thus, we can represent $\boldsymbol{f}$ as $\boldsymbol{f} = (f_{X_1}, f_{X_2})^T$. To simplify the notation, we can represent it as $\boldsymbol{f} = (f_1, f_2)^T$. In Figure 18, we can observe the specific learned $\boldsymbol{f}^t$ of single-learner teaching, vanilla and communicated MINT. Based on the observations from Figure 18, it can be concluded that MINT method is much more effective in terms of efficiency when compared to single-learner teaching. Furthermore, for communicated MINT, the teacher can provide the communicated matrix to let $\boldsymbol{f}$ being learnt within a single iteration. Besides, Figure 19 shows the loss plot which tracks convergence performance.

**Synthetic 1D data.** We use synthetic one-dimensional data to evaluate when the communication matrix $A_t$ is advantageous. Specifically, our main focus is on $t = 1$, which refers to the first iteration

---

[7]http://athenasc.com/Bivariate-Normal.pdf

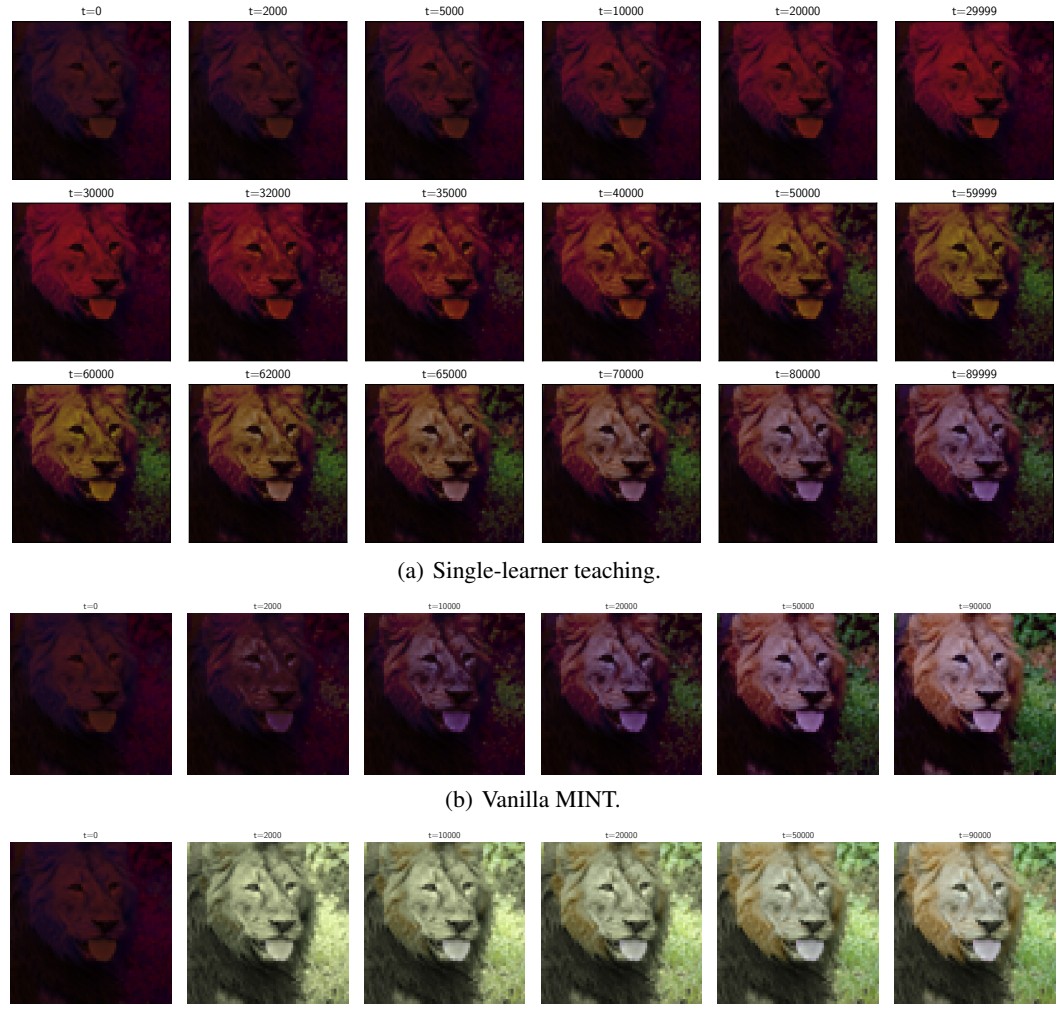

(a) Single-learner teaching.

(b) Vanilla MINT.

(c) Communicated MINT.

Figure 11: Visualization of $\boldsymbol{f}^t$ taught by GFT under a particular initialization.

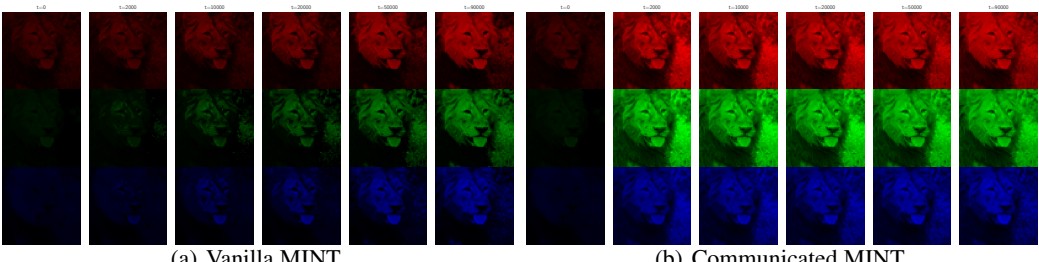

(a) Vanilla MINT.  (b) Communicated MINT.

Figure 12: The channel-wise visualization of specific $\boldsymbol{f}^t$ corresponding to Figure 11 (b)-(c). The communicated MINT exhibits better performance compared to vanilla MINT.

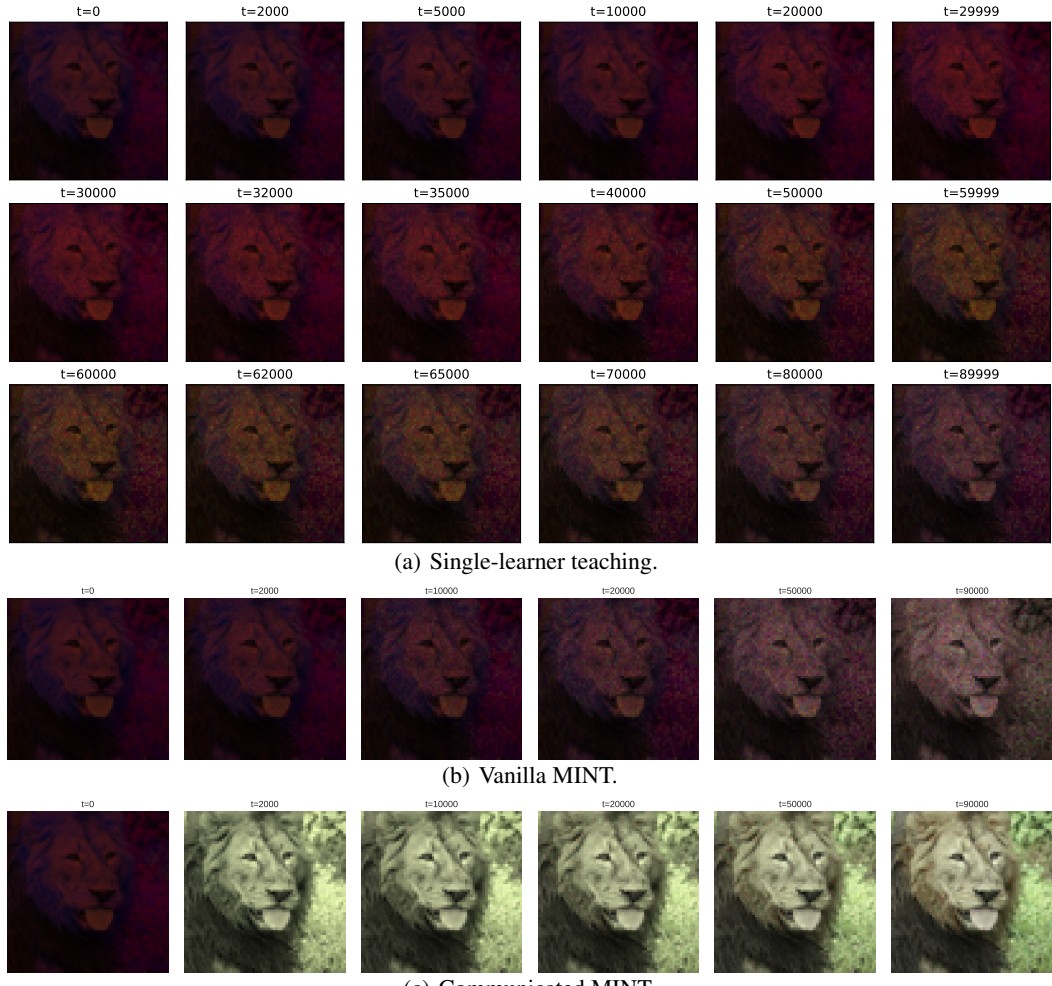

(a) Single-learner teaching.

(b) Vanilla MINT.

(c) Communicated MINT.

Figure 13: Visualization of $\boldsymbol{f}^t$ taught by RFT under a particular initialization.

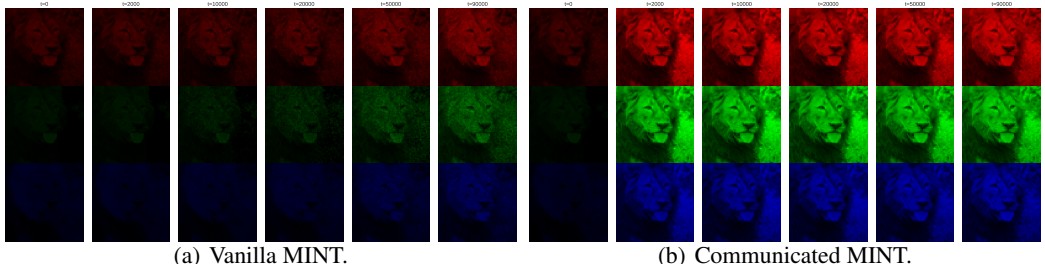

(a) Vanilla MINT.

(b) Communicated MINT.

Figure 14: The channel-wise visualization of specific $\boldsymbol{f}^t$ corresponding to Figure 13 (b)-(c).

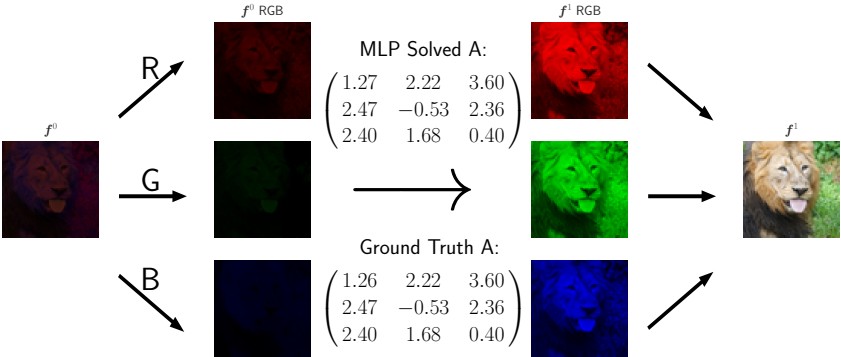

Figure 15: Illustration of an extreme case of teaching. The teacher can help the learners directly learn $\boldsymbol{f}^*$ through providing the communication matrix.

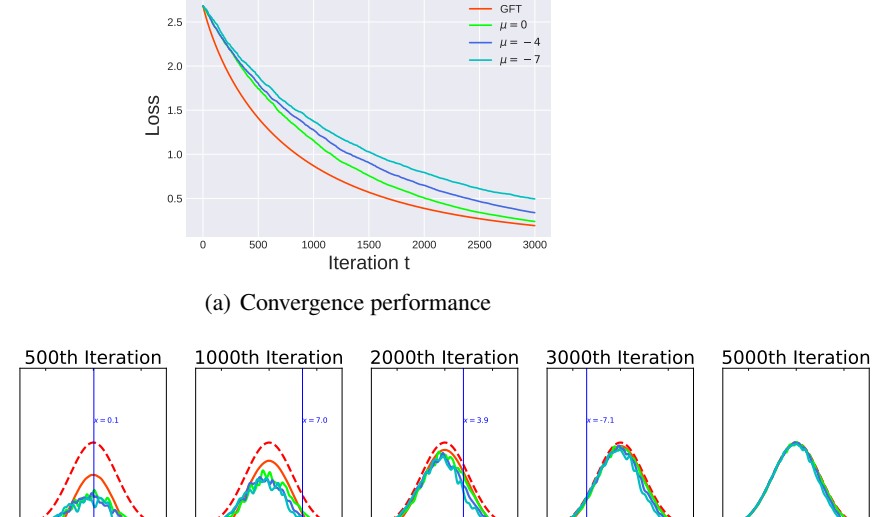

(a) Convergence performance

(b) Visualization of $f^t$ and $f^*$

Figure 16: Examination of relation between RFT and GFT.

(the results of $t > 1$ can be derived similarly), where there are two learners attempting to learn separate target functions. We set $\boldsymbol{f}^0 = (f_1, f_2)^T = (x, x^2)^T$ and $\boldsymbol{f}^*_\gamma = (\gamma x/2 + \gamma x^2/2 + (1 - \gamma)\cos x, \gamma x/3 + 2\gamma x^2/3 + (1 - \gamma)\sin x)^T, \gamma \in [0, 1]$, and we generate $\gamma$ by `arange(-1,1,0.01)`. In Figure 20, we present the corresponding $\|A_\gamma \boldsymbol{f}^0 - \boldsymbol{f}^*_\gamma\|_{\mathcal{H}^2}$ against $\gamma$. The observed trend is that as $\gamma$ increases, the distance between $A_\gamma \boldsymbol{f}^0$ and $\boldsymbol{f}^*_\gamma$ decreases. This indicates that $A_\gamma$ performs better when $\boldsymbol{f}^*$ can be linearly expressed by $\boldsymbol{f}^0$, but poorly when $\boldsymbol{f}^*$ cannot be linearly expressed by $\boldsymbol{f}^0$. Moreover, $A$ is beneficial when $\boldsymbol{f}^*$ can be partially expressed in a linear manner by $\boldsymbol{f}^0$.

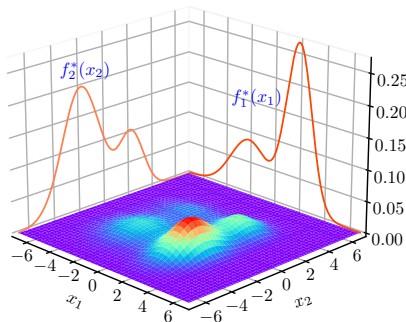

Figure 17: The visualization of a multivariate mixture Gaussian distribution $f^*_{X_1,X_2}(x_1, x_2)$.

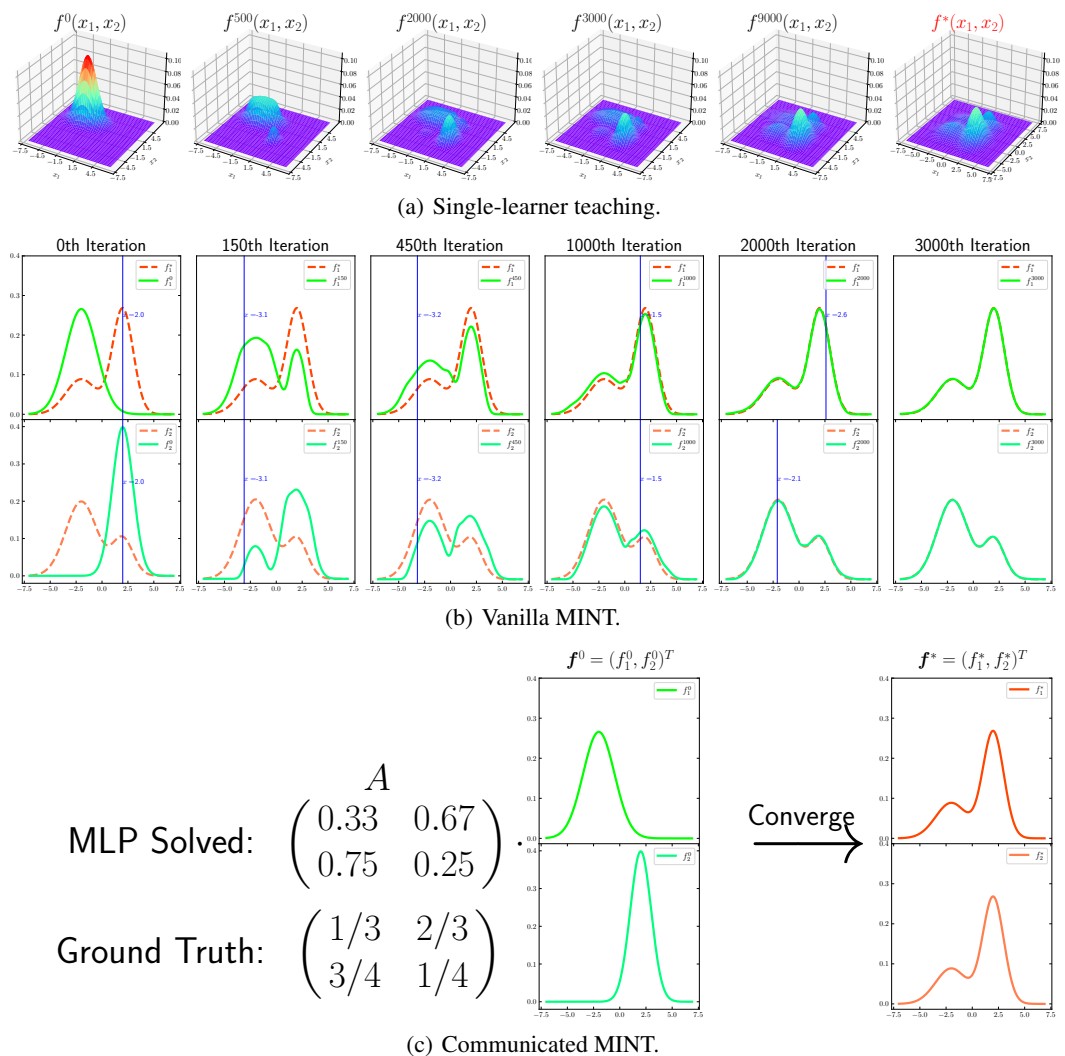

(a) Single-learner teaching.

(b) Vanilla MINT.

(c) Communicated MINT.

Figure 18: The specific learned $\boldsymbol{f}^t = (f_1^t, f_2^t)^T$.

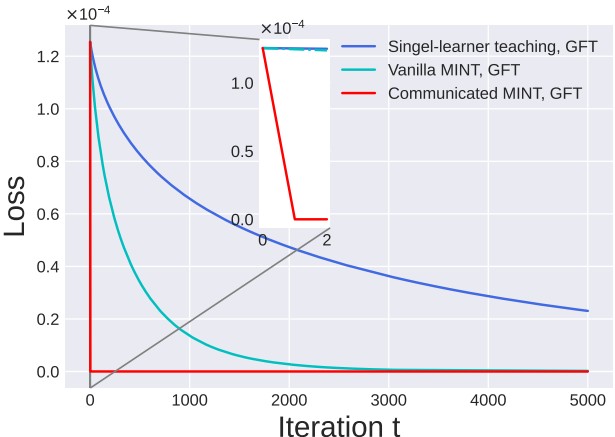

Figure 19: Convergence performance for the synthetic bivariate mixture Gaussian data.

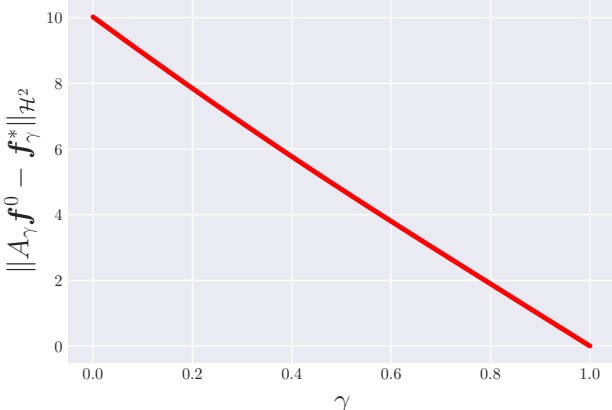

Figure 20: The distance between $A_\gamma \boldsymbol{f}^0$ and $\boldsymbol{f}_\gamma^*$ VS. $\gamma$.

