# OpenReview forum: "Nonparametric Teaching for Multiple Learners"
_NeurIPS.cc/2023/Conference — NeurIPS 2023 poster_

### Official Review · Reviewer_dq11 · 2023-07-05

**Soundness:** 3 good
**Presentation:** 3 good
**Contribution:** 3 good
**Rating:** 7
**Confidence:** 1

**Summary:**

This paper extends nonparametric teaching from the setting of teaching each learner independently to teaching multiple ones simultaneously. The method is about teaching a vector-valued model, which improves over existing methods when multiple learners can communicate with each other. There are both theoretical and experimental results validating the effectiveness of this method.

**Strengths:**

The paper seems to have solid theoretical analysis; it's hard for me to judge as a non-expert.

**Weaknesses:**

Only one experiment on the RGB channels of images is shown in the paper. Another experiment in a different setting may help demonstrate the generalizability of the approach. As someone with no expertise in this area, I'll defer to others on the technical side of things.

**Questions:**

Can you elaborate on what other possible applications can benefit from the proposed approach?

---

> ### Author Rebuttal · Authors · 2023-08-09
>
> We thank the reviewer for the positive comments! We respond in detail to your specific concerns in the following.
>
> **Q1**: Due to the page limitation, we have provided two experiment results in the main paper. We also show additional performance evaluations of MINT under various settings and presented detailed demonstrations in the appendix. These include testing MINT with a specific initialization of $f^0$ in RGB teaching tasks, comparing RFT with GFT using channel-wise visualization, and conducting experiments with Synthetic Bivariate Mixture Gaussian data. All of these experiments align with our theoretical findings that highlight the superior efficiency of communicated MINT over the vanilla approach, which in turn outperforms single-learner teaching. We will improve our presentation to note our additional experiments in appendix.
>
> **Q2**: This theoretical work has the potential for application in the field of knowledge distillation [g], where the teacher (cumbersome) model is to transfer knowledge to the learner (small) model by sharing “soft targets”. The idea presented in this work could serve as inspiration for future investigations into distilling the knowledge of a teacher to multiple learners. Such a knowledge transferring framework, in turn, can open up new avenues of exploration and possibilities across different domains, such as computer vision [h], Internet-of-Things [i], natural language processing [j] and decision-making tasks [k]. In the revision, we will expand upon these potential applications by providing additional discussions to delve deeper into their implications.
>
> [g] Hinton et al. Distilling the knowledge in a neural network. NeurIPS 2014 Deep Learning Workshop.
> [h] Wang et al. Gradient-based algorithms for machine teaching. CVPR 2021.
> [i] Xu et al. Locality sensitive teaching. NeurIPS 2021.
> [j] Li et al. Blip: Bootstrapping language-image pre-training for unified vision-language understanding and generation. ICML 2022.
> [k] Yengera et al. Curriculum Design for Teaching via Demonstrations: Theory and Applications. NeurIPS 2021.

---

### Official Review · Reviewer_eAYw · 2023-07-05

**Soundness:** 3 good
**Presentation:** 2 fair
**Contribution:** 2 fair
**Rating:** 4
**Confidence:** 3

**Summary:**

The paper extends non-parametric machine teaching to the case of multiple learners. In particular, each learner learns one component of a vector-valued function. The authors consider the case where learners have no communication with each other and the case where there is "communication" via a matrix transformation of the outputs of each individual learner.

**Strengths:**

Originality: I am not an expert in machine teaching so I can't say how original the contribution is relative to prior work.

Quality: The paper appears to be of good quality, though I did not check the proofs carefully.

Clarity: The paper was relatively clear, though with key exceptions referenced below

Significance: The paper appears to me to be an incremental step beyond the non-parametric single-learner setting.

**Weaknesses:**

I believe the paper needs more motivation and needs to contrast with the case where a single learner is learning a vector-valued function. Currently, the paper extends the scalar-valued single-learner setting to one where the teacher is teaching multiple learners---each outputting one component of a vector-valued output. In the introduction, the paper explains why teaching multiple learners at once would be better than teaching multiple learners separately. That seems quite clear, but then why not just teach one single learner that is learning a vector-valued function rather than multiple learners learning each component separately? The introduction seems to hint at computational issues but this should be made more explicit, and if computational issues are a critical motivation, then experiments showing the computational advantage should be included.

Especially because the paper also includes a section on "communication" between the learners and shows that they achieve better performance when communication is allowed. (By the way, the communication, in this case, is just a matrix that the teacher gives each learner.. the learners aren't learning to communicate with each other, so it seems clear from the outset that this would lead to improvement) So if communication leads to higher performance, then why not full "communication", i.e., one learner that outputs all components?

**Questions:**

I thought the experiments section could be made clearer (I also looked at the additional details in the appendix but was still confused). What is the domain and range of the functions being learned? What are the examples (x and y) that the teacher is giving for each settting? Are they pixels?

**Limitations:**

I ask that the authors replace the Lenna image with any other image. There is no reason that this paper needs to use, of all images, the Lenna image, a cropped photo of a nude women from a Playboy centerfold. The continued unnecessary use of this image perpetuates an uninclusive environment.

https://womenlovetech.com/losing-lena-why-we-need-to-remove-one-image-and-end-techs-original-sin/
https://www.washingtonpost.com/opinions/a-playboy-centerfold-does-not-belong-in-tj-classrooms/2015/04/24/76e87fa4-e47a-11e4-81ea-0649268f729e_story.html

Lena herself has said that she no longer wants her image used: “I retired from modelling a long time ago,” said Lena in a new documentary film called Losing Lena. “It’s time I retired from tech, too. We can make a simple change today that creates a lasting change for tomorrow. Let’s commit to losing me.”

---

> ### Author Rebuttal · Authors · 2023-08-09
>
> Thanks for the useful comments. We are deeply appreciative of the reviewer’s efforts to help us improve our paper. We take all comments seriously and try our best to address every raised concern. We sincerely hope that our response can resolve your concerns. Any follow-up questions are welcome.
>
> **Q1**: Our motivation for teaching multiple learners, with each learner focusing on learning a separate component of a vector-valued function, stems from the need to align with the scenario described in [c], where a single learner is taught a scalar-valued target model or function. Additionally, teaching multiple learners is an important problem under exploration in machine teaching [a-b]. Interestingly, the mathematical framework of analysis for teaching a single learner that learns a vector-valued function and teaching multiple learners, each focusing on learning a specific component, should be essentially consistent. Both approaches address the question of how to effectively teach a vector-valued function under the framework of vector-valued functional optimization. The difference lies in the multi-learner approach, where each learner is connected to a specific component of the vector-valued function, whereas in the case of a single learner, the entire vector-valued function is connected to that learner.
>
> In this work, “computationally wasteful” is used to describe that a larger number of iterations are required to achieve convergence for single-learner teaching than multi-learner one. We also illustrate the benefits of multi-learner teaching over single-learner teaching in terms of the loss plot, as demonstrated in figures such as Fig.3 and Fig.10. We will make a clearer presentation in the revision.
>
> **Q2**: “one learner outputs all components” may be a setting in the case where a single learner learns a vector-valued function and it makes sense within that context. Since the setting in this work is aligned with that in [c] where a single learner is taught a scalar-valued target function and our focus in this work is on teaching multiple learners, communication occurs among multiple learners. From a mathematical perspective, these two forms of communication should be essentially consistent since they both involve analyzing the relationships among the components of vector-valued target functions.
>
> **Q3**: In line 322, we provide an explanation that a grayscale image can be visualized as a three-dimensional surface, where the z-axis represents the level of gray, and the x and y axes indicate the pixel coordinates [c]. The domain is determined by the x and y values, which represent the pixel locations, while the range is represented by the z values, indicating the gray levels. In the case of RGB images, each color channel can also be visualized as a three-dimensional surface, similar to grayscale images. Again, the domain corresponds to the pixel locations, and the range represents the corresponding color values. As for Synthetic 1D Gaussian data, the domain is [-14,14], while the range corresponds to the values generated by the Gaussian distribution $\mathcal{N}(x; 0, 5^2)$. We will further polish the presentation in the revision.
>
> **Q4**: The Lenna image used in Figure 1 and the experiments conducted in this theoretical work is merely an example of an RGB image, which we consider to be a well-known image in computer vision. As our attention mainly lies in theoretical machine learning community, we acknowledge that there is a possibility of us not being fully informed about all the news concerning Lenna. We apologize for any concerns caused by our lack of awareness regarding recent news about Lenna. Rest assured, we will replace the image with a different RGB image in the revision, while ensuring that it does not impact the theoretical findings presented in this work.

---

> > ### Comment · Reviewer_eAYw · 2023-08-16
> >
> > Thank you for the response. Unfortunately, I don't think this rebuttal addressed my concerns with the motivation as the response mainly referenced the fact that others are working on it, e.g. "stems from the need to align with the scenario described in [c]" or "additionally, teaching multiple learners is an important problem under exploration in machine teaching [a-b]", but still does not provide a motivation for the scenario in the first place.

---

> > > ### Author Response · Authors · 2023-08-18
> > >
> > > Thanks a lot for the additional response. We apologize for not being clear enough. We are more than happy to address them. We sincerely hope that our response can address your concerns.
> > >
> > > Indeed, such two cases are similar, and they are two different standpoints of considering the question of how to effectively teach a vector-valued function under the framework of vector-valued functional optimization.
> > >
> > > Teaching multiple learners presents a more flexible scenario where the teaching for each learner can conclude upon learning a component of the vector-valued target function. In contrast, teaching a single learner who learns the entire vector-valued function can only terminate once all components of the vector-valued target function are learnt, with the efficiency being determined by the worst-case scenario.
> > >
> > > Besides, the multi-learner setting offers a general framework that can be generalized to more complicated scenarios. For instance, the cases where each learner operates within a different feature space would be impractical with a single-learner setting that teaches a single learner who is learning a vector-valued function. Specifically, this multi-learner framework enables to capture the diversity, as each individual learner is responsible for learning a component of the vector-valued target function, a capability that is limited in a single-learner setting. On the other hand, when the feature space is consistent across all learners, these two cases become interchangeable, demonstrating that the single-learner setting is a special case within the broader multi-learner setting.
> > >
> > > In practice, teaching multiple learners is a very common scenario where the interplay and trade-off among multiple learners are taken into consideration. In this setting, the efficiency of teaching multiple learners needs to be studied -- whether the convergence speed-up from the single-learner teaching holds is particularly important. More broadly, as optimal education is one of the most important motivations for machine teaching, multi-learner teaching brings the machine teaching research closer to the reality.

---

> > > ### Author Response · Authors · 2023-08-21
> > >
> > > We sincerely thank you for your comments on our submission. We have taken great care to address each comment in detail in our rebuttal.
> > >
> > > Just a warm reminder that the discussion period is drawing to a close on Aug 21st at 1 pm EDT. We would greatly appreciate it if you could acknowledge receipt of our further responses and inform us whether we have addressed your concerns. We are eager to engage in any further discussions if needed. Once again, thank you for your valuable feedback.

---

> > > > ### Comment · Area_Chair_kVtq · 2023-08-22
> > > >
> > > > well received your response which be considered seriously. thanks.

---

### Official Review · Reviewer_JyMg · 2023-07-06

**Soundness:** 4 excellent
**Presentation:** 4 excellent
**Contribution:** 4 excellent
**Rating:** 6
**Confidence:** 4

**Summary:**

The paper studied nonparametric teaching in the presence of multiple learners. Following prior works on nonparametric teaching, the paper extended to a scenario where multiple learners simultaneously learn a separate component of the joint model. The paper first analyzed the performance of the Random Functional Teaching (RFT) and the Greedy FT (GFT) strategy and showed that both teachers can successfully guarantee reduction in the loss function when the learners are completely independently, and no communication happens. Secondly, the authors studied the effect of communication between learners and show that an affine transformation of the joint model does not increase the loss but can significantly enhance the loss reduction in the beginning of the training phase. Experiments validated the discoveries in this paper.

**Strengths:**

(1). The problem of teaching multiple learners itself is interesting and underexplored in the machine teaching community. This paper pushes the frontier of machine teaching in this aspect.

(2). The paper theoretically analyzed the loss reduction of RFT and GFT teacher, and also the benefit of communication. The results show that non-parameter teaching with communication can indeed help with loss reduction.

(3). The paper performed extensive empirical study of the teaching strategy, and the results are convincing.

**Weaknesses:**

(1). The assumption in the theoretical study is not clearly stated in the paper. For example, the theory seems to rely on the fact that the loss function must be convex, and this information should be made more prominent upfront. The authors may want to discuss the applicability of both the theory and the methodology developed in this paper. Does it apply with neural network model or only convex learners?

(2). One thing missing from the discussion is that how does the teaching performance compare to vanilla learning. The non-parametric teacher only guarantees that the loss is always reduced. However, how does the loss reduction compare to standard learning process without teaching is an important topic to be discussed.

**Questions:**

(1). What is the applicability of the theoretical results and the methodology developed in this paper? Is convexity a required property?

(2). How does non-parametric teaching compare to vanilla learning without teaching. Does it help accelerate the learning process?

**Limitations:**

Yes

---

> ### Author Rebuttal · Authors · 2023-08-09
>
> Thanks for the encouraging comments. We sincerely thank the reviewer's efforts for helping us improve the paper. We hope that our response resolves your concerns.
>
> **Q1**: Thanks for pointing it out. We introduce the convex loss assumption above Eq.(7), and in the revision, we will make sure to highlight prominent assumptions earlier. Currently, our results are derived based on convex learners, which serves as a crucial step towards non-convex cases (e.g., neural networks). And exploring the convergence conditions for neural networks is an intriguing avenue to pursue. For instance, investigating the condition under which neural networks converge to critical points [f] could involve employing non-convex optimization techniques.
>
> Regarding applicability, one potential application of this work could be in the field of knowledge distillation [g], where the teacher (cumbersome) model is to transfer knowledge to the learner (small) model by sharing “soft targets”. The idea in this study may serve as inspiration for future research on distilling a teacher's knowledge to multiple learners. Such knowledge transferring paradigms, in turn, can offer new insights and possibilities in various domains, including computer vision [h], Internet-of-Things [i], natural language processing [j] and decision-making tasks [k]. In the revised version, we will provide additional discussions to further explore these potential applications.
>
> [f] Diakonikolas et al. Sever: A robust meta-algorithm for stochastic optimization. ICML 2019.
> [g] Hinton et al. Distilling the knowledge in a neural network. NeurIPS 2014 Deep Learning Workshop.
> [h] Wang et al. Gradient-based algorithms for machine teaching. CVPR 2021.
> [i] Xu et al. Locality sensitive teaching. NeurIPS 2021.
> [j] Li et al. Blip: Bootstrapping language-image pre-training for unified vision-language understanding and generation. ICML 2022.
> [k] Yengera et al. Curriculum Design for Teaching via Demonstrations: Theory and Applications. NeurIPS 2021.
>
> **Q2**: In essence, random functional teaching (RFT) employs a random sampling strategy and serves as a straightforward baseline that can be considered as the functional counterpart of stochastic gradient descent. As it randomly selects examples, RFT can also be seen as a form of "learning without teaching". On the other hand, the greedy functional teaching (GFT) teacher adopts a greedy approach by selecting examples that maximize the gradient. Through theoretical and empirical analysis, we have demonstrated that GFT outperforms RFT in terms of efficiency. In the revision, we will provide additional explanations to better illustrate these points.

---

> > ### Comment · Reviewer_JyMg · 2023-08-20
> >
> > Thank you for your response. My questions are addressed in the rebuttal. The paper is technically solid and above average, so I decided to keep my current positive score.

---

### Official Review · Reviewer_D1DQ · 2023-07-30

**Soundness:** 3 good
**Presentation:** 3 good
**Contribution:** 2 fair
**Rating:** 6
**Confidence:** 3

**Summary:**

This paper investigates the iterative machine teaching problem under the non-parametric learner setting with vector-valued target models, also known as multi-learner nonparametric teaching (MINT). The authors consider two teaching strategies: Random Functional Teaching (RFT) and Greedy FT (GFT). The authors first theoretically analyze the convergence behavior induced by the teaching strategies in the vanilla MINT setting, where there is no communication between the learners. Then, they study the teaching strategies in the communicated MINT setting. Finally, they empirically compare the effectiveness of different teaching strategies.

**Strengths:**

The paper is overall well-written, and the related work is extensively discussed.

The theoretical results in this paper seem correct; I haven’t checked the details of the proofs.

**Weaknesses:**

The availability of a powerful teacher with a vector-valued target model needs to be motivated with more realistic practical scenarios.

The novelty of the contributions of this work in comparison to “Nonparametric Iterative Machine Teaching (NIMT)”: the problem formulation/setup, RFT/GFT teaching strategies are extensions from the NIMT paper. Thus, it is important to clearly discuss how non-trivial these extensions are and how different the proof techniques are from those of the NIMT paper.

**Questions:**

The theoretical results in the paper are based on a synthetic example generation setting. If the teacher is restricted to choosing examples from a pool, that would result in different examples than the ideal one. Then, what would be the impact on the results?

Given that the teacher can freely synthesize examples to guide the learner toward a target model, can the results discussed be extended to learners with non-convex loss functions?

**Limitations:**

The paper is of an algorithmic/theoretical nature and does not have any direct potential negative societal impact.

---

> ### Author Rebuttal · Authors · 2023-08-09
>
> Thanks for the useful comments. We are deeply appreciative of the reviewer’s efforts to improve our paper. We take all comments seriously and try our best to address every raised concern. We sincerely hope that our response resolves your concerns.
>
> **Q1**: An important problem towards realistic application in machine teaching lies in classroom teaching [a-b], where a teacher is responsible for teaching multiple learners. One motivation behind this work comes from exploration of such realistic multi-learner teaching, and we conduct investigation of multi-learner teaching based on nonparametric teaching. Specifically, we generalize the model space from space of scalar-valued functions to that of vector-valued functions. We will add more discussion on realistic practical scenarios in the revision.
>
> [a] Yeo et al. Iterative classroom teaching. AAAI 2019.
> [b] Zhu et al. No learner left behind: on the complexity of teaching multiple learners simultaneously. IJCAI 2017.
>
> **Q2**:
> - In terms of contribution, we extend single-learner nonparametric teaching [c] to general multi-learner one. We achieve this by formulating multi-learner teaching as teaching of vector-valued functions under the framework of vector-valued functional optimization. Considering the correlation between the components of a vector-valued function has the potential to enhance the efficiency of teaching, we also investigate communicated teaching scenarios where multiple learners can execute linear combination on the currently learnt functions of all learners, which is more practical and non-trivial.
>
> - From a technical standpoint, we explain the difference in proof techniques between this work and single-learner teaching [c] in line 219 and 245. Specifically, we introduce the expectation operation over random sampling, which allows us to average out the impact of randomness. This also enables us to quantify the difference between RFT and GFT by introducing the distance between $ {x^t_i}^∗$ and $\mu_i$ in Theorem 10, line 280, which is not considered in [c].
>
> - In experiments, we conduct investigations in more comprehensive multi-learner teaching settings (e.g., RGB images and bivariate mixture gaussian data), demonstrating the effectiveness of multi-learner teaching and expanding the potential applications of nonparametric teaching.
>
> We will further polish the presentation to highlight these in the revision.
>
> [c] Zhang et al. Nonparametric iterative machine teaching. ICML 2023.
>
> **Q3**: The teaching ability of pool-based teachers is limited due to their constrained knowledge domain, which is a subset of that of synthesis-based teachers. This constraint can result in the learner converging to a suboptimal ${f^*}’$, as mentioned briefly in line 190. We will add more discussion in the revision.
>
> **Q4**: Very inspiring question! In this work, the analysis of the loss function is conducted under the assumption of convexity, which serves as a stepstone towards handling nonconvex scenarios. It would be interesting to theoretically investigate the convergence performance and additional conditions of convergence for non-convex loss, as they may vary depending on the specific task at hand.
>
> For instance, we can potentially treat a non-convex loss locally as a convex one [d], enabling us to apply the methodology developed in this work straightforwardly. By doing so, we can analyze the local convergence globally. Additionally, it might be a potential extension to make use of convex relaxation technique [e] to transform a non-convex problem into a convex one for handling nonconvex problems.
>
> [d] Razaviyayn et al. Parallel successive convex approximation for nonsmooth nonconvex optimization. NeurIPS 2014.
> [e] Xie et al. Orthogonality-promoting distance metric learning: convex relaxation and theoretical analysis. ICML 2018.

---

> > ### Comment · Reviewer_D1DQ · 2023-08-21
> >
> > I thank the authors for their response and for addressing my concerns.

---

### Decision · Program_Chairs · 2023-09-21

**Decision:**

Accept (poster)

**Comment:**

The paper studied nonparametric teaching under the multiple learners setting. The paper extended the prior work to a scenario where multiple learners simultaneously learn a separate component of the joint model. The paper analyzed two approaches (no communication between learners): Random Functional Teaching (RFT) and the Greedy FT (GFT) strategy, with theoretical guarantee of reduction in the loss function when the learners are completely independently. On the top of that, the authors studied the effect of communication between learners and show that an affine transformation can significantly enhance the loss reduction in the beginning of the training phase. Empirical studies were also provided to validate the theoretical analysis. This paper studies an interesting/important problem in machine teaching and the theoretical results are sound and with valuable insights. We recommend to accept this paper. But in the meantime, this paper can be better motivated not just simply citing some other work, especially why not just teach one single learner that is learning a vector-valued function rather than multiple learners learning each component separately?